# The ADAM17 sheddase complex regulator iTAP/Frmd8 modulates inflammation and tumor growth

Marina Badenes[1,2,3] , Emma Burbridge[1,4], Ioanna Oikonomidi[1], Abdulbasit Amin[1,5], Érika de Carvalho[1,6] , Lindsay Kosack[1], Camila Mariano[1] , Pedro Domingos[6] , Pedro Faísca[1], Colin Adrain[1,4]

The metalloprotease ADAM17 is a sheddase of key molecules, including TNF and epidermal growth factor receptor ligands. ADAM17 exists within an assemblage, the "sheddase complex," containing a rhomboid pseudoprotease (iRhom1 or iRhom2). iRhoms control multiple aspects of ADAM17 biology. The FERM domain–containing protein iTAP/Frmd8 is an iRhom-binding protein that prevents the precocious shunting of ADAM17 and iRhom2 to lysosomes and their consequent degradation. As pathophysiological role(s) of iTAP/Frmd8 have not been addressed, we characterized the impact of iTAP/Frmd8 loss on ADAM17-associated phenotypes in mice. We show that iTAP/Frmd8 KO mice exhibit defects in inflammatory and intestinal epithelial barrier repair functions, but not the collateral defects associated with global ADAM17 loss. Furthermore, we show that iTAP/Frmd8 regulates cancer cell growth in a cell-autonomous manner and by modulating the tumor microenvironment. Our work suggests that pharmacological intervention at the level of iTAP/Frmd8 may be beneficial to target ADAM17 activity in specific compartments during chronic inflammatory diseases or cancer, while avoiding the collateral impact on the vital functions associated with the widespread inhibition of ADAM17.

## Introduction

The release of signaling proteins from the plasma membrane often involves the so-called "shedding": the proteolytic cleavage of the extracellular domains of integral membrane proteins (Lichtenthaler et al, 2018), catalyzed by membrane-tethered metalloproteases. This process of combining a membrane-anchored substrate and a membrane-tethered protease enables the strict spatiotemporal control over signaling that is required during development, homeostasis, and immune responses. ADAM17 is a prominent sheddase that cleaves/releases a range of important signaling molecules including the apical inflammatory cytokine TNF, activating ligands of the epidermal growth factor receptor (EGFR), and key cell adhesion molecules including L-selectin (Zunke & Rose-John, 2017). ADAM17 substrates control multiple important biological processes including inflammation, growth control, development, metabolism, and cell adhesion (Zunke & Rose-John, 2017).

In recognition of the importance of ADAM17 for inflammatory disease and cancer, there have been numerous attempts by Pharma to develop safe ADAM17 inhibitors (Calligaris et al, 2021). However, these attempts have failed at (or before) phase II trials because of safety concerns caused by collateral targeting of other members of the wider metzincin metalloprotease subgroup that share a common active-site architecture with ADAM17 (Murumkar et al, 2020). New approaches are therefore needed to identify strategies to safely block the cleavage of ADAM17 substrates during disease, ideally targeting a specific tissue or compartment.

Over the past 10 yr, significant knowledge has emerged concerning how ADAM17 is regulated at the biochemical, cell biological, and organismal levels. We and others discovered that certain pseudoenzymes called iRhoms are essential co-factors for ADAM17; without iRhoms, ADAM17 is proteolytically inactive, and hence, ADAM17 substrate shedding is abolished (Adrain et al, 2012; McIlwain et al, 2012; Siggs et al, 2012; Li et al, 2015; Adrain & Cavadas, 2020). Mammals encode two iRhom paralogs with partially redundant roles in ADAM17 regulation at the organismal and cellular levels (Christova et al, 2013; Li et al, 2015). iRhom1 regulates ADAM17 shedding in the nervous system (Sun et al, 2021; Tüshaus et al, 2021) and in endothelial cells (Babendreyer et al, 2020). iRhom2 KOs develop normally but fail to secrete TNF, a key ADAM17 substrate that coordinates the responses to infection and chronic inflammatory diseases; loss of iRhom2 attenuates the development of multiple inflammatory diseases in mouse models (Adrain et al, 2012; McIlwain et al, 2012; Siggs et al, 2012; Issuree et al, 2013; Luo et al, 2016; Barnette et al, 2018; Chaohui et al, 2018; Qing et al, 2018; Sundaram et al, 2019; Adrain & Cavadas, 2020; Kim et al, 2020). In contrast, the double KO of iRhom1 and iRhom2 in mice results in embryonic or perinatal lethality (Christova et al, 2013; Li et al, 2015).

[1]Instituto Gulbenkian de Ciência, Oeiras, Portugal   [2]Faculty of Veterinary Medicine, Lusofona University, Lisbon, Portugal   [3]Faculty of Veterinary Nursing, Polytechnic Institute of Lusofonia, Lisbon, Portugal   [4]Patrick G Johnston Centre for Cancer Research, Queen's University, Belfast, UK   [5]Department of Physiology, Faculty of Basic Medical Sciences, University of Ilorin, Ilorin, Nigeria   [6]Instituto de Tecnologia Química da Universidade Nova de Lisboa (ITQB-Nova), Oeiras, Portugal

Correspondence: c.adrain@qub.ac.uk
Ioanna Oikonomidi's present address is Genentech, South San Francisco, CA, USA

This tissue-specific regulation of ADAM17 by different iRhom proteins has appealing therapeutic implications, because if it was possible to specifically target an individual iRhom (e.g., iRhom2), this would avoid the severe collateral effects associated with global inhibition of ADAM17 activity. Hence, targeting iRhom2 may enable specific inhibition of the inflammatory properties of ADAM17, while avoiding the global inhibition of its activity.

iRhom proteins physically interact with ADAM17, forming an assemblage called the "sheddase complex" (Künzel et al, 2018; Oikonomidi et al, 2018), and fulfill many key roles in ADAM17 regulation. This includes the vesicular transport of ADAM17 from its site of biogenesis, the ER, into the *trans*-Golgi (Adrain et al, 2012; McIlwain et al, 2012), where ADAM17 undergoes a crucial activation step: the removal of its inhibitory prodomain by proprotein convertases (Schlöndorff et al, 2000). Therefore, ADAM17 is catalytically inactive in iRhom KO cells because it fails to exit the ER and cannot undergo this critical maturation step (Adrain et al, 2012). Follow-up studies from several groups later found that ADAM17 remains complexed to iRhom proteins on the cell surface, where iRhoms act as a signal-sensing scaffold within the sheddase complex (Cavadas et al, 2017; Grieve et al, 2017). Upon receipt of suitable signaling cues (e.g., phorbol esters, inflammatory stimuli), the cytoplasmic tail of iRhom proteins undergoes phosphorylation, which drives the recruitment of 14-3-3 proteins to this domain of iRhom (Cavadas et al, 2017; Grieve et al, 2017). This event triggers the activation of ADAM17, potentially driven by triggering a molecular rearrangement of the complex or displacing ADAM17 from the sheddase complex (Adrain & Cavadas, 2020).

To understand in more detail how the sheddase complex is regulated, we and others performed immunoprecipitation experiments coupled to mass spectrometry to isolate iRhom-binding partners. This identified a largely uncharacterized protein that we named iTAP (iRhom tail–associated protein), also called Frmd8 (Künzel et al, 2018; Oikonomidi et al, 2018). iTAP/Frmd8 is a member of the ERM (ezrin, radixin, moesin) superfamily, whose canonical function is anchoring membrane protein "clients" to cortical actin at the plasma membrane (Chishti et al, 1998; Fehon et al, 2010). Consistent with this general role of FERM domain proteins, iTAP/Frmd8 binds to the cytoplasmic tail of iRhom proteins and it is essential to maintain the stability of the sheddase complex on the plasma membrane (Künzel et al, 2018; Oikonomidi et al, 2018). iTAP/Frmd8 ablation triggers the selective loss of mature (active) ADAM17 in a range of iTAP/Frmd8 KO cells and mouse tissues, because of precocious mis-sorting of the sheddase complex to lysosomes where it is degraded (Künzel et al, 2018; Oikonomidi et al, 2018).

Like iRhom2 KOs, iTAP/Frmd8 KO mice are born at normal Mendelian ratios, appear normal, and are fertile (Künzel et al, 2018; Oikonomidi et al, 2018). In the present study, we sought to characterize the impact of loss of iTAP/Frmd8 on ADAM17-associated phenotypes using the iTAP/Frmd8-null mice generated by our laboratory (Oikonomidi et al, 2018). Our data suggest that iTAP/Frmd8 KO mice do not exhibit the collateral defects associated with global loss of ADAM17 but exhibit defects in inflammatory and intestinal epithelial barrier functions that have been reported in ADAM17 mutant models. Furthermore, we show that iTAP/Frmd8 regulates cancer cell growth in a cell-autonomous manner and by modulating the tumor microenvironment. Our work identifies that

intervention at the level of iTAP/Frmd8 may be beneficial to target the inflammatory features of ADAM17 associated with iRhom2, and shows new insights of the iTAP/Frmd8-mediated sheddase complex on cancer growth regulation.

# Results

## iTAP/Frmd8 KO mice exhibit loss of mature ADAM17 in multiple tissues and reduced shedding of the ADAM17 substrate L-selectin

To understand better the role of iTAP/Frmd8 as a regulator of the sheddase complex, we and others previously generated iTAP/Frmd8-null mice (Künzel et al, 2018; Oikonomidi et al, 2018). *Frmd8* homozygous mutant mice are born at normal Mendelian ratios, are viable, are fertile, and do not present any gross anatomical abnormalities compared with WT littermates (Künzel et al, 2018; Oikonomidi et al, 2018). However, as these previous studies did not exclude the possibility of subtle phenotypes, we carried out a detailed characterization of iTAP/Frmd8 KO mice. In some, but not all (Li et al, 2007; Gelling et al, 2008), ADAM17 KO mouse studies, ADAM17-null mice exhibit perinatal lethality and pronounced epithelial defects in multiple tissues, many of which are associated with defective EGFR signaling (Peschon et al, 1998), including precocious eyelid opening, defects in eye, lung, and heart valve morphogenesis, and deranged hair follicles with irregular pigment deposition (Peschon et al, 1998). Many of these phenotypes are also observed in iRhom1/iRhom2 double KO (DKO) embryos (Li et al, 2015).

However, aside from defective inflammatory responses, iRhom2-null mice only present defects in keratinocyte proliferation, presenting thinner footpads and irregular pigment deposition (Maruthappu et al, 2017). We found that iTAP/Frmd8 KO embryos develop normally and exhibit no obvious morphological differences in any of the major organs surveyed (Fig S1A). During mouse embryonic development, the eyelids of WT mice undergo an ADAM17/EGFR-dependent epithelial fusion step (Peschon et al, 1998). WT mice are born with the appearance of their eyes "closed," after which eyelids open 11–12 d after birth. In contrast, ADAM17-deficient embryos are born with their eyes precociously open because of a failure in epithelial eyelid fusion (Peschon et al, 1998). Like iRhom2 KOs, whose eyelids develop normally, probably because of redundancy with iRhom1, iTAP/Frmd8 KOs lacked ADAM17-associated defects in the eye, which were indistinguishable from WT eyes at E14.5 (Fig S1B) and after birth. Other epithelial tissues (small intestine, colon, cecum, lung, and the heart [Fig S1C and D], respectively) also exhibited no obvious differences compared with controls. We conclude that iTAP/Frmd8-null mice undergo normal epithelial and cardiac development.

Deletion of iTAP/Frmd8 triggers the loss of the lower molecular weight form of ADAM17 that corresponds to the mature (i.e., active) form of the protease (Schlöndorff et al, 2000), as a consequence of its precocious trafficking to and degradation in lysosomes (Künzel et al, 2018; Oikonomidi et al, 2018). Interestingly, isolated primary keratinocytes from iTAP/Frmd8 KOs exhibited substantially reduced levels of mature ADAM17 (Fig S1E and F); however, some residual amounts of mature ADAM17 remained (Fig S1E and F). These

residual levels of mature ADAM17 may potentially explain the lack of overt histological skin or hair follicle density defects in iTAP/Frmd8 KOs (Fig S1G–I). Finally, as a hypermetabolic phenotype has been reported for the minority of ADAM17 KOs that survive perinatal lethality until adulthood (Gelling et al, 2008), we tested the growth rate and food intake of iTAP/Frmd8 KO juveniles. Although this does not preclude differences upon metabolic challenge, iTAP/Frmd8 KO mice appeared normal under steady-state conditions (Fig S1J and K). In summary, like iRhom2 KO mice but unlike ADAM17 KO or iRhom1/iRhom2 DKO animals, iTAP/Frmd8 KOs do not present overt growth or developmental defects.

As our previous studies established that ablation of iTAP/Frmd8 in primary human macrophages results in a pronounced impairment of TNF shedding by ADAM17 (Oikonomidi et al, 2018), here we focused on the physiological impact of iTAP/Frmd8 loss on the protein levels and sheddase activity of ADAM17 in immune cells of mice, in which ADAM17 and iRhom2 play a prominent role (Horiuchi et al, 2007; Li et al, 2007; Adrain et al, 2012; McIlwain et al, 2012; Siggs et al, 2012). Strikingly, we observed that iTAP/Frmd8-deficient splenocytes and lymphocytes exhibited substantially reduced levels of mature and total ADAM17 (Fig 1A and B). We also observed substantial depletion of mature and total ADAM17 in B cells isolated from spleen or lymph nodes, and in dendritic cells differentiated ex vivo from bone marrow progenitors (Fig S2A–C; gating strategy for B-cell sorting indicated in Fig S3A–E). To further probe the impact of iTAP/Frmd8 depletion on ADAM17 activity in immune cells, we assessed the shedding of the endogenous ADAM17 substrate L-selectin (Marczynska et al, 2014). L-selectin is a prominent cell adhesion molecule expressed on most leukocytes and plays a role in the migration of neutrophils and other immune cells to sites of inflammation (Ivetic et al, 2019). L-selectin is shed upon immune cell activation; ADAM17 appears to be the major L-selectin–shedding protease because thymocytes and other leukocytes lacking ADAM17 fail to shed L-selectin upon their activation by the phorbol ester PMA (Peschon et al, 1998; Li et al, 2006).

Notably, we found that the PMA-triggered shedding of L-selectin was substantially impaired in T cells from iTAP/Frmd8 KO spleens (Fig 1C–E; gating strategy indicated in Fig S4A–H) and to a more modest extent in B cells (Fig 1F; gating strategy indicated in Fig S4A–D and I). Consistent with a role of defective ADAM17 activity, L-selectin shedding could be rescued by the broad-spectrum metalloprotease inhibitor marimastat but was insensitive to the ADAM10-selective inhibitor GI 254023X (Fig 1C–F). In summary, together with previous studies (Künzel et al, 2018; Oikonomidi et al, 2018), our data indicate that iTAP/Frmd8 is a physiological regulator of mature ADAM17 levels and of ADAM17 sheddase activity in immune cells.

### iTAP/Frmd8 KOs exhibit iRhom2 defects in regulating STING and the innate immune response to HSV-1 infection

Recently, it was shown that iRhom2 is essential for the activity of STING (Luo et al, 2016), a central player in innate immune responses to DNA viruses, such as HSV-1. Accordingly, iRhom2 deficiency impairs the immune response (e.g., induction of interferon-associated genes) to HSV-1 in vitro and iRhom2 deficient mice are more prone to HSV-1 infection (Luo et al, 2016). To determine the involvement of iTAP/Frmd8 in this pathway, we compared the protein levels of STING in

uninfected iTAP/Frmd8 KO versus iRhom2 KO mouse–derived embryonic fibroblasts (MEFs) relative to WT controls. This revealed that as reported for iRhom2 KOs, iTAP KO MEFs have reduced levels of STING protein, although the extent of the reduction was less marked than in iRhom2 KO cells (Fig 2A and B). In addition, we found that the mRNA levels of the immunomodulators il-6 and Ifnb1, which are induced by the STING pathway, were constitutively down-regulated in iTAP/Frmd8 KO cells as in iRhom2 KO MEFs, both under steady-state conditions (Fig 2C and D) and in response to HSV-1 infection (Fig 2E and F). The secreted protein level of IFN-β was also decreased mostly in response to HSV-1 infection (Fig 2G). In contrast, the mRNA levels of the HSV-1 genes UL-42 and ICP22 were unchanged between groups (Fig 2H and I), indicating that HSV-1 could infect all three genotypes similarly.

### iTAP/Frmd8 KOs exhibit ADAM17 defects in experimental sepsis and colitis models

As noted above, iRhom2 plays an essential role in the trafficking regulation and sheddase activity of ADAM17 in immune cells, particularly macrophages—a key TNF source in vivo, the major inflammatory cytokine whose release requires the ADAM17/iRhom2 sheddase complex (Adrain et al, 2012; McIlwain et al, 2012; Siggs et al, 2012). To discern the requirement for iTAP/Frmd8 for inflammatory cytokine release, we challenged iTAP/Frmd8 KOs with two inflammatory models. We found that in response to LPS stimulation, bone marrow–derived macrophages differentiated ex vivo from iTAP/Frmd8 KO mice exhibited substantially less TNF shedding than WT counterparts, whereas, as expected from iRhom KO studies (Adrain et al, 2012), there was no alteration in the levels of the soluble cytokine IL-6, whose release does not require ADAM17 (Fig 3A and B). IL-6 transcription/secretion in response to LPS is driven by the TLR4 pathway, not STING, explaining the intact IL-6 response in this setting. To probe this observation in an in vivo context, we challenged iTAP/Frmd8 KO mice in a sepsis model driven by i.p. administration of LPS (Fig 3C and D). Similar to the in vitro studies, we found that iTAP KO mice have significantly less serum TNF, but as anticipated, unaltered IL-6 levels compared with controls (Fig 3C and D). Hence, similar to iRhom2 KO or ADAM17 mutant mice, iTAP/Frmd8 controls the shedding of TNF in vivo. Although ADAM17 and iRhom2 are required for the propagation of inflammatory responses, the predominant role of ADAM17 in the intestinal epithelium appears to be the promotion of epithelial repair (Chalaris et al, 2010). Hence, ADAM17 mutants are sensitized to intestinal damage driven by an experimental dextran sulfate sodium (DSS)–induced colitis model (Chalaris et al, 2010). Consistent with this ADAM17 phenotype, iTAP/Frmd8 KO mice were more prone to worsened indicators of disease triggered by a model of acute DSS-induced colitis (Fig 3E), including more pronounced body weight loss (Fig 3F) and colon shortening (Fig 3G). Associated with this, iTAP/Frmd8 KOs exhibited more severe disease in a histopathological-based score of disease severity (IBD score) (Seamons et al, 2013), particularly in the mid-colon, with more pronounced mucosal loss and more area of the intestine affected (Fig 3H–J). To address the effect of iTAP/Frmd8 loss on epithelial repair capacity after the DSS insult, we induced colitis but let the mice recover for 5 d after DSS exposure (Fig S5A). This established that the KO mice could not recover body weight to the same extent as the WT mice (Fig S5B), had a shorter colon (Fig S5C), and had more severe IBD in the histopathological analysis, specifically in the mid-colon (Fig S5D–F). Our

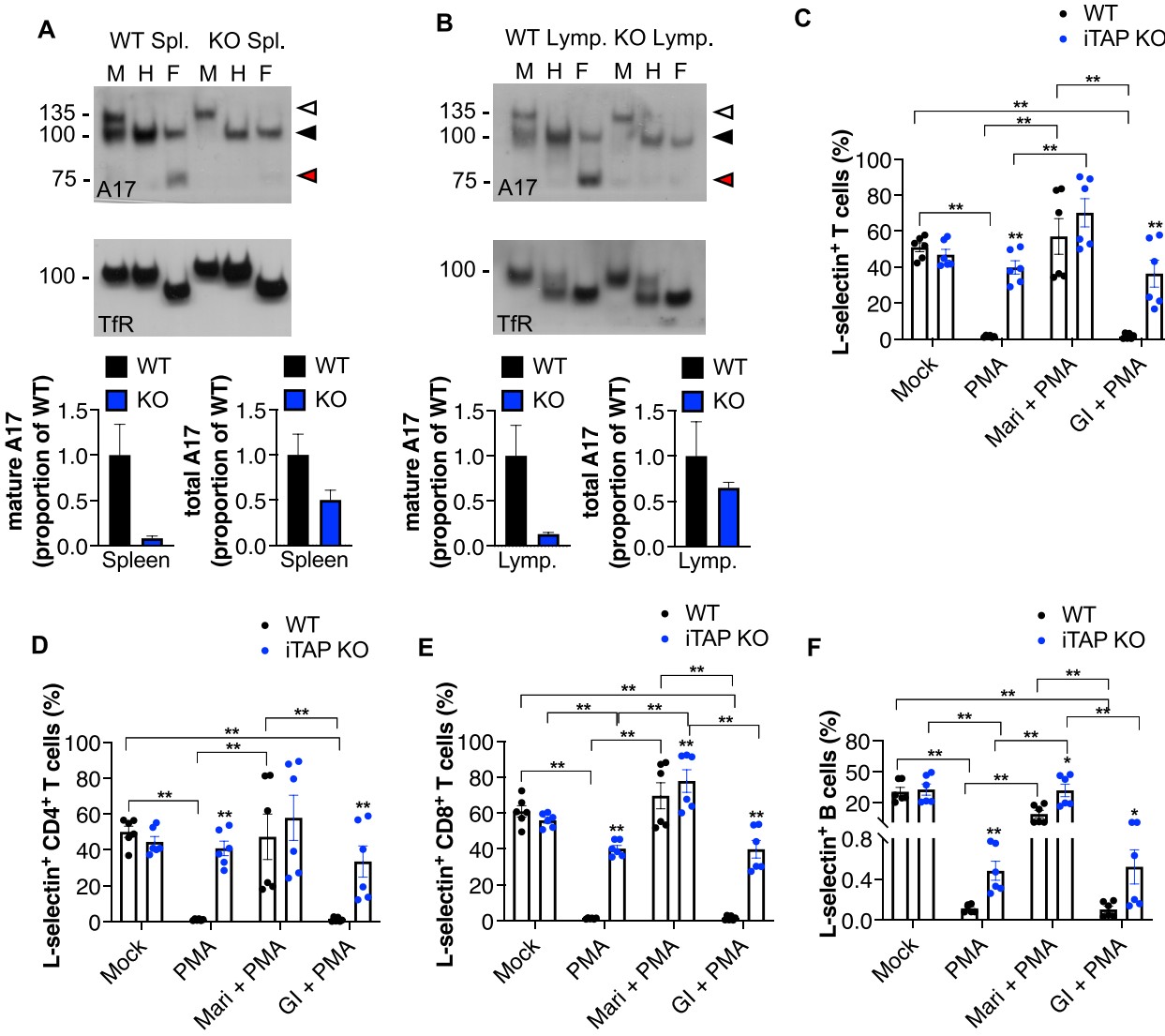

**Figure 1. iTAP/Frmd8 KO mice have defects in ADAM17 maturation and shedding.**
**(A, B)** Immunoblots showing immature versus mature ADAM17 (A17) and respective densitometric analysis of mature and total ADAM17 in splenocytes (A) and lymphocytes (B) from WT versus iTAP/Frmd8 KO mice. Protein samples were either mock-treated (M) or deglycosylated with Endo-H (H) or PNGase F (F). Immunoblots for the transferrin receptor serve as a loading control. n = 2. The immature form of ADAM17 is indicated by white arrowheads; the black arrowhead denotes both glycosylated mature ADAM17 and deglycosylated immature ADAM17, respectively (which have similar electrophoretic mobility), whereas red arrowheads denote the fully deglycosylated, mature, ADAM17 polypeptide. **(C, D, E, F)** Levels of L-selectin in the surface of T cells (CD45.2$^+$, CD3$^+$) (C), CD4-positive T cells (CD45.2$^+$, CD3$^+$, CD4$^+$) (D), CD8-positive T cells (CD45.2$^+$, CD3$^+$, CD8$^+$) (E), and B cells (CD45.2$^+$, B220$^+$) (F) isolated from the spleen of the mice described above. The isolated splenocytes were pretreated with DMSO, 5 μM ADAM17 inhibitor marimastat (Mari), or 1 μM ADAM10 inhibitor GI254023X (GI) and then treated with DMSO or 1 μM PMA. n = 2 with three technical replicates per experiment. Throughout: results are indicated as the mean ± SEM. The Mann–Whitney–Wilcoxon test was used; * represents $P < 0.05$, and ** represents $P < 0.01$.

data hence establish that like ADAM17, loss of iTAP/Frmd8 exacerbates experimental colitis in vivo and impairs the capacity of recovery after the insult.

## iTAP/Frmd8 regulates tumor growth from within the tumor microenvironment

ADAM17 is crucial for the shedding of multiple substrates that can promote a state of chronic inflammation, contributing to the development and/or dissemination of cancer, such as TNF, or a soluble form of the IL-6 receptor (Miller et al, 2017; Düsterhöft et al, 2019). Tumor cells often present increased ADAM17 expression leading to autocrine EGFR activation (Dong et al, 1999; Borrell-Pagès et al, 2003; Gschwind et al, 2003; Schumacher & Rose-John, 2022). In addition, ADAM17 through TNF/TNFR1 signaling is crucial for tumor cell–induced endothelial cell necroptosis and vascular permeability, which facilitates tumor cell extravasation and metastasis (Bolik et al, 2022). As ADAM17 maturation is decreased in multiple tissues of iTAP/Frmd8 KO mice, including the lung (Oikonomidi et al, 2018), and defects in its sheddase activity were observed in ex vivo (Figs 1C–F and 3A) and in in vivo (Fig 3C) experiments, we next evaluated the effect of iTAP/Frmd8 depletion on tumor growth and

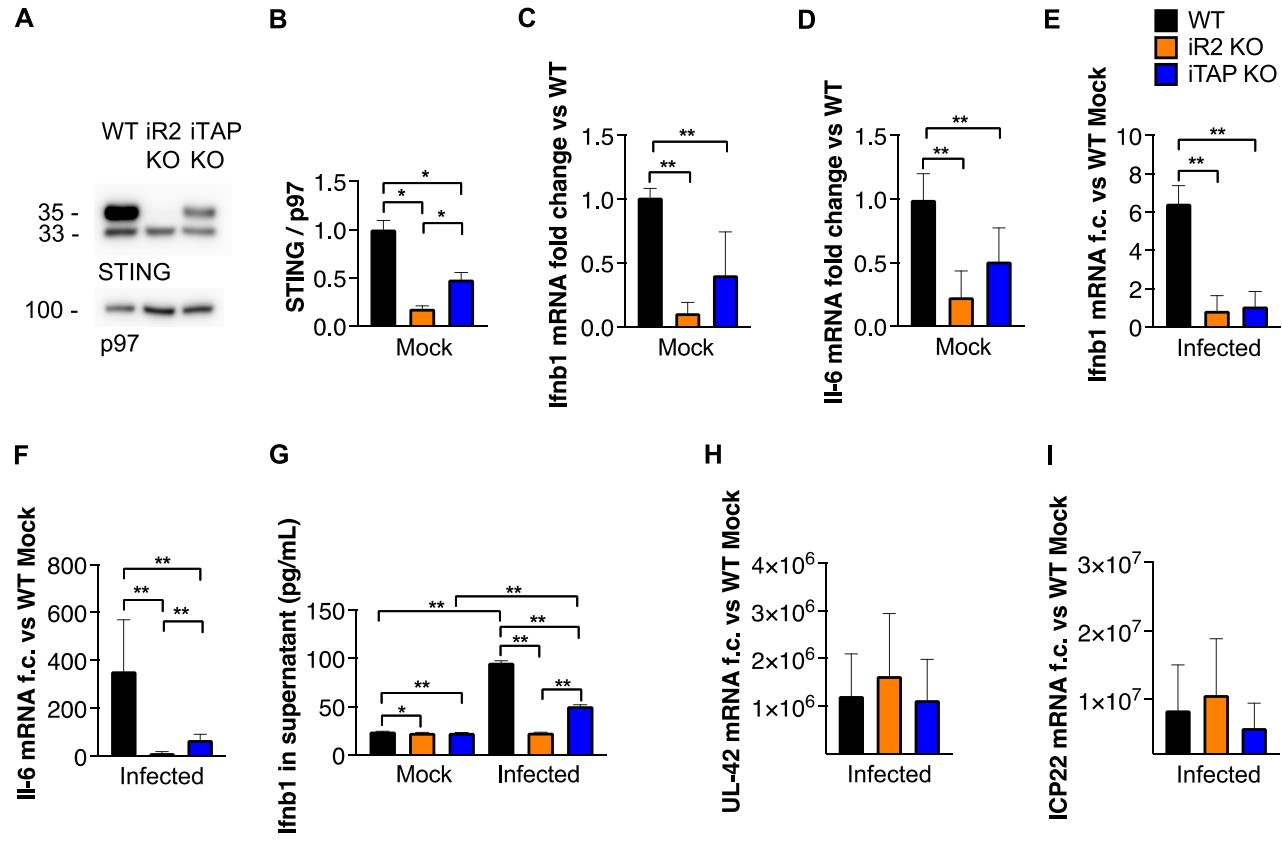

**Figure 2. iTAP/Frmd8 KO MEFs have reduced levels of STING protein and impaired STING immune responses.**
**(A, B)** Immunoblot (A) and respective densitometric analysis (B) showing the level of STING protein expression in MEFs from WT versus iRhom2 and iTAP/Frmd8 KO mice. Immunoblot for p97 is used as a loading control. **(C, D)** mRNA expression of *Ifnb1* (C) and *il-6* (D) in non-infected iRhom2 and iTAP/Frmd8 KO MEFs shown as fold change (f.c.) relative to their level in WT cells. **(E, F)** mRNA expression of *Ifnb1* (E) and *il-6* (F) in HSV–1–infected iRhom2 and iTAP/Frmd8 KO MEFs relative to WT mock cells. **(G)** Levels of IFN-β protein in the supernatant of non-infected and HSV-1-infected cells. **(H, I)** Relative mRNA expression of HSV-1 viral *UL-42* (H) and *ICP22* (I) transcripts in infected cells. n = 2 with three technical replicates per experiment. Throughout: results are indicated as the mean ± SEM. The Mann–Whitney–Wilcoxon test was used; * represents $P < 0.05$, and ** represents $P < 0.01$.

metastasis. We used an allograft model based on the subcutaneous injection of murine Lewis lung carcinoma (LLC) tumor cells into immunocompetent WT versus iTAP/Frmd8 KO mice (Fig 4A), to assess the impact of iTAP/Frmd8 loss in the tumor niche on the growth of WT LLC cells (Mendonça et al, 2019). Strikingly, we found that iTAP/Frmd8 KO host mice were protected from tumor growth, resulting in tumors of a lower volume and mass (Fig 4B–E). Consistent with a role of iTAP/Frmd8 in the dissemination of inflammatory responses, we observed smaller necrotic lesions (Fig 4F) and reduced superficial ulceration (data not shown) in tumors from iTAP/Frmd8 KO hosts, and decreased levels of the mRNAs for inflammatory cytokines (*TNF*, *Il-6*, and *Il-1β*) and chemokines (*CXCL2* and *CCL4*) compared with tumors from WT mice (Fig S6A). In this model of subcutaneous-mediated LLC cell inoculation, we did not detect significant metastases to the lungs (a site where LLC cells typically accumulate in other models). However, we detected reduced *TNF* and *Il-1β* mRNA levels in the lungs of the KOs compared with controls (Fig S6B), suggesting that loss of iTAP/Frmd8 may result in a less inflammatory milieu. Taken together, our data reveal that iTAP/Frmd8 expression in the tumor microenvironment promotes tumor growth.

## Cell-autonomous expression of iTAP/Frmd8 promotes tumor cell growth

ADAM17 is overexpressed in many human cancers, including lung cancer, whereas genetic or pharmacological ablation of ADAM17 suppressed tumor proliferation and dissemination partly by promoting EGFR ligand release (McGowan et al, 2008; Jiao et al, 2018; Saad et al, 2019a, 2019b; Ni et al, 2020; Schumacher & Rose-John, 2022). Having observed that iTAP/Frmd8 expression in recipient host mice affects WT LLC tumor cell growth in vivo, we next assessed whether cell-autonomous iTAP/Frmd8 expression influences LLC tumor cell growth, this time in WT immunocompetent recipient mice (Fig 5). To this end, we generated iTAP/Frmd8 KO LLC cells (KO LLC) via CRISPR, and LLC cells overexpressing iTAP/Frmd8. We verified that the LLC KO cells did not differ from the parental cells in terms of other alterations besides iTAP/Frmd8 depletion, by analyzing the level of multiple proteins from different compartments (Fig S7A–M). In pairwise experiments, we confirmed that the levels of *iTAP/Frmd8* gene expression and of the mature and total amounts of ADAM17 were decreased in iTAP/Frmd8 KO cells compared with control cells (parental cells expressing Cas9) (Figs

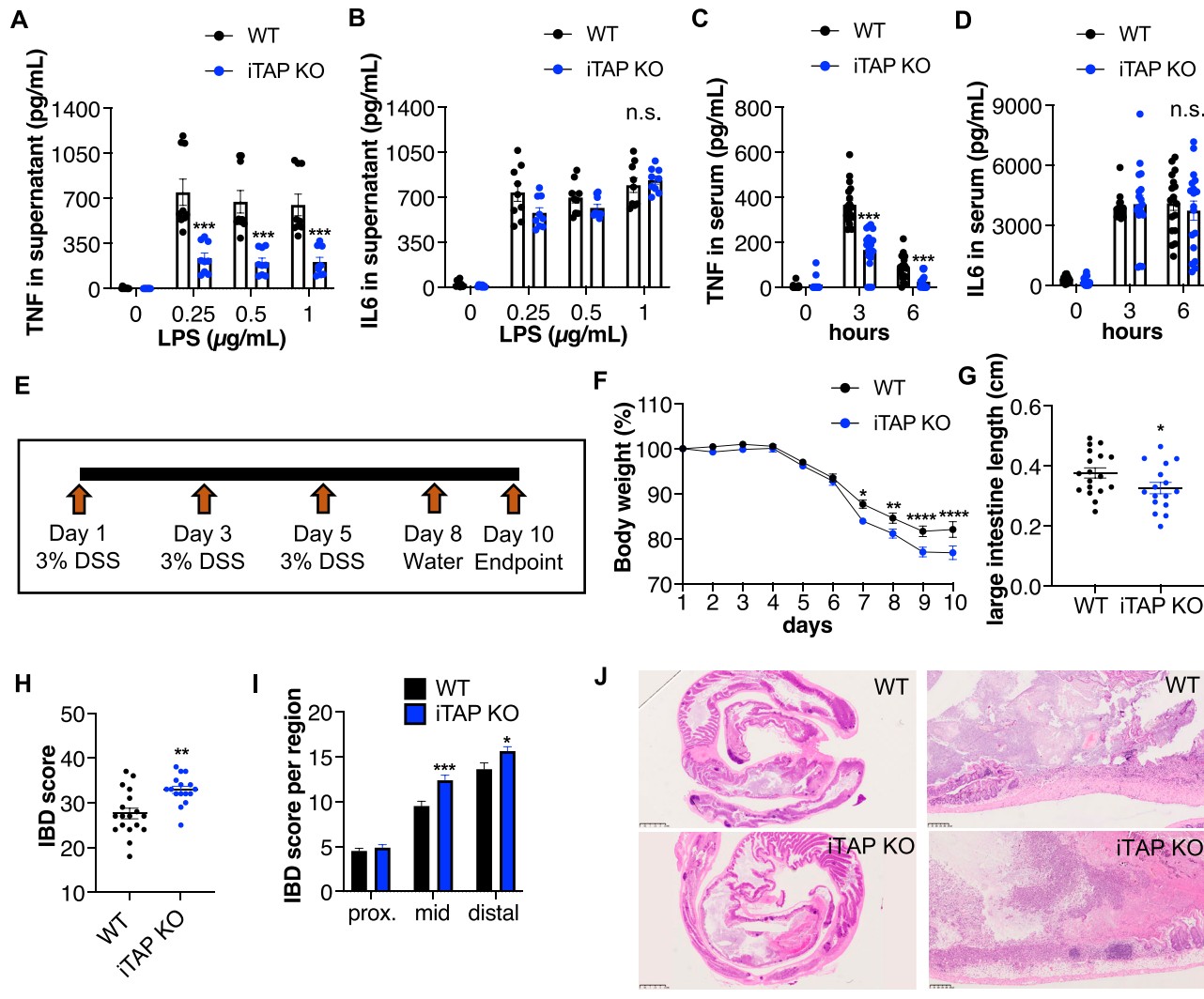

**Figure 3. iTAP/Frmd8 KO mice have ADAM17-associated defects in TNF shedding and are more susceptible to colitis.**
**(A, B)** Level of TNF (A) and IL-6 (B) secretion in culture supernatants from primary macrophages isolated from WT versus iTAP/Frmd8 KO mice and stimulated with three doses of LPS (0.25, 0.5, and 1 μg/ml) for 2 h (TNF) or 6 h (IL-6). n = 3 with three replicates per genotype per experiment. **(C, D)** TNF (C) and IL-6 (D) levels in the serum of WT versus iTAP/Frmd8 KO mice after *i.p.* injection of PBS (0) or LPS (37.5 μg/g) for 3 or 6 h. n = 3 with six mice per genotype per condition. **(E)** Schematics of the DSS-driven acute colitis model. **(F, G)** Body weight (%) (F), and large intestine length (cm) (G) of WT versus iTAP/Frmd8 KO mice after DSS administration. **(H, I, J)** Histopathological IBD score in the total area (H) or by section (proximal, mid, and distal) (I), and H&E images of the large intestine (J) of the mice described above. n = 3 with six mice per genotype per experiment. Results are indicated as the mean ± SEM. A scale bar is indicated within the H&E images. The Mann–Whitney–Wilcoxon test was used throughout. For the analysis in (F), two-way ANOVA was used. * represents $P < 0.05$, ** represents $P < 0.01$, *** represents $P < 0.001$, and **** represents $P < 0.0001$.

5A and B and S7N) and that iTAP/Frmd8 overexpression compared with cells that were transduced with the respective retroviral empty vector enhanced *iTAP/Frmd8* gene and protein expression and ADAM17 maturation and the overall levels of ADAM17 (Figs 4C and D and S7O and P). Before animal studies, we carried out an in vitro characterization of these cell lines to test whether the modulation of iTAP/Frmd8 levels in LLC cells could impact upon the shedding of the ADAM17 substrates TNF, HB-EGF, and amphiregulin, which could potentially promote differences in cell proliferation and/or migration in in vivo experiments. We did not find detectable levels of secreted TNF or HB-EGF (data not shown), but interestingly, the level of amphiregulin shedding by LLC cells was high (Fig 5E and F). We found that when the cells were stimulated with PMA, to promote

ADAM17 activation, iTAP/Frmd8 KO cells shed significantly less amphiregulin (Fig 5E), whereas the opposite was found in iTAP/Frmd8-overexpressing cells (Fig 5F). This effect appeared dependent on ADAM17 because the secretion of amphiregulin in WT cells was significantly decreased by the metalloprotease inhibitor BB-94, but not by the ADAM10-selective inhibitor GI254023X (Fig 5E and F). Our data establish that iTAP/Frmd8 regulates the shedding of endogenous EGFR ligands.

We next tested the impact of these cells when injected subcutaneously into immunocompetent WT mice (Fig 5G), finding that the local tumors in the mice that received the iTAP/Frmd8 KO cells had a reduced growth rate, leading to smaller tumor volume and mass compared with parental controls (Fig 5H–J). The inverse was

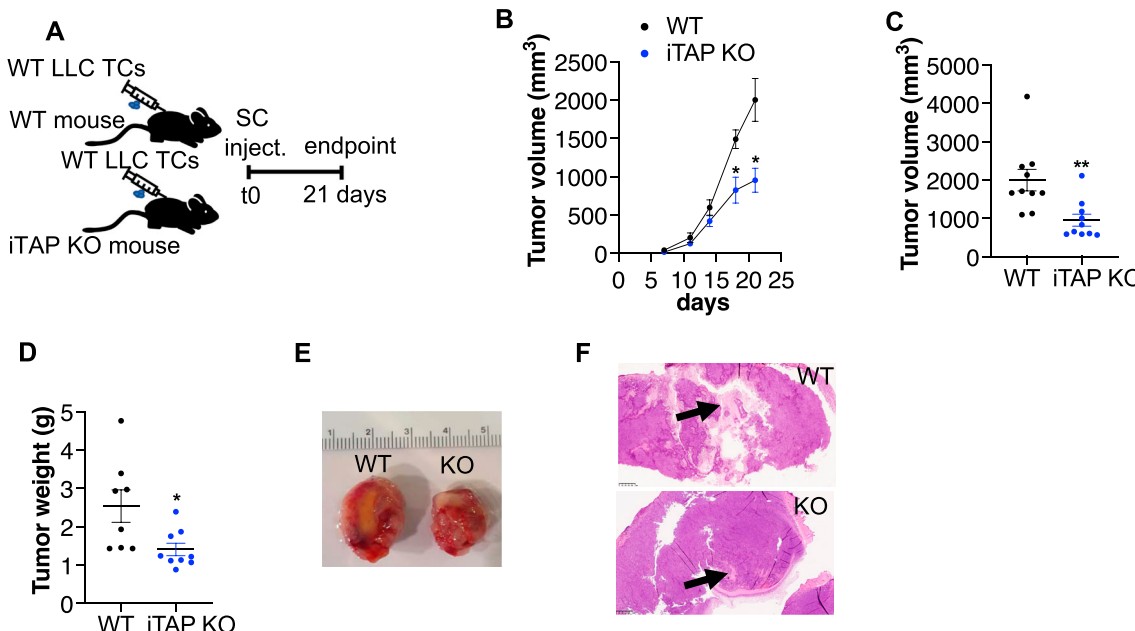

**Figure 4. iTAP/Frmd8 expression in the microenvironment induces tumor growth.**
**(A)** Schematic illustrating the tumor model based on subcutaneous injection of WT LLC cells into WT versus iTAP/Frmd8 KO immunocompetent hosts. **(B, C, D, E)** LLC cell–derived tumor volume over time (B) and at the endpoint (C), tumor weight (D), and representative photographs of the tumors (E) isolated from WT and iTAP/Frmd8 KO mice. **(F)** Images of necrotic areas of the tumors described above. Results are indicated as the mean ± SEM. A scale bar is indicated within the H&E images. The Mann–Whitney–Wilcoxon test was used; * represents $P < 0.05$, ** represents $P < 0.01$, and *** represents $P < 0.001$.

observed in the mice that received the iTAP/Frmd8-overexpressing LLC cells compared with vector controls (Fig 5H–J). In support of these findings, iTAP/Frmd8 KO tumors had fewer proliferating cells, as assessed by Ki67 staining (Fig 5K and L) and *Pcna* mRNA levels in the tumors (Fig S8); the opposite was found in iTAP/Frmd8-overexpressing tumors (Figs 5K and L and S8).

### Cell-autonomous iTAP/Frmd8 expression promotes tumor cell dissemination

As ADAM17 overexpression is associated with a higher risk of metastasis and a poorer prognosis in several types of cancer (McGowan et al, 2008; Jiao et al, 2018; Ni et al, 2020), we next determined the cell-autonomous impact of iTAP/Frmd8 upon tumor emergence in the lungs after the intravenous injection into WT mice of iTAP/Frmd8 KO LLC cells versus their appropriate parental control line (Fig 6A–E). In separate pairwise experiments, we also compared the injection of empty vector–bearing versus iTAP/Frmd8-overexpressing cells (Fig 6F–J). In this model, loss of iTAP/Frmd8 significantly reduced the number of lung tumors (Fig 6B), and the tumor volume (Fig 6C) and burden (Fig 6D) compared with parental WT cells. Consistent with this, although not statistically significant, the opposite trend was observed when comparing empty vector–transduced cells versus their respective iTAP/Frmd8-overexpressing cells (Fig 6F–J).

As our results (Figs 5 and 6A–J) indicate that iTAP/Frmd8 can promote LLC tumor cell growth and proliferation, we turned to simpler in vitro models to explore this effect. We confirmed in vitro in pairwise experiments that the iTAP/Frmd8 KO LLC cells

proliferate more slowly than parental control cells, whereas iTAP/Frmd8-overexpressing cells proliferated more rapidly than their equivalent empty vector–transduced cells. (Fig 6K and L). Taken together with the experiments shown in Figs 4 and 5, our data establish that iTAP/Frmd8 influences tumor growth in two ways: in a cell-autonomous manner that appears to involve control of cellular proliferation, and a cell–non-autonomous manner, by influencing the tumor niche.

## Discussion

Studies on iTAP/Frmd8, performed by our laboratory and others, revealed that this molecule binds to iRhom proteins and is crucial to stabilize the sheddase complex on the cell surface, avoiding its mis-sorting to and degradation in lysosomes (Künzel et al, 2018; Oikonomidi et al, 2018). However, the pathophysiological role(s) of iTAP/Frmd8 were not fully addressed until now. Our study identifies iTAP/Frmd8 as an in vivo regulator of ADAM17: in addition to regulating the levels of mature and total ADAM17 in multiple tissues (Oikonomidi et al, 2018) (Figs 1, S1, and S2), iTAP/Frmd8 KO mice exhibit phenotypes associated with ADAM17-shedding defects in several immune cell compartments. This includes reduced shedding of the ADAM17 substrate L-selectin (Marczynska et al, 2014) from T lymphocytes and B cells (Fig 1) and impaired LPS-triggered release of TNF from macrophages ex vivo and in vivo (Fig 3), which depends on ADAM17 (Bell et al, 2007; Horiuchi et al, 2007). Shedding defects have been reported for iRhom2 mutant mice, which also develop normally, but exhibit pronounced inflammatory

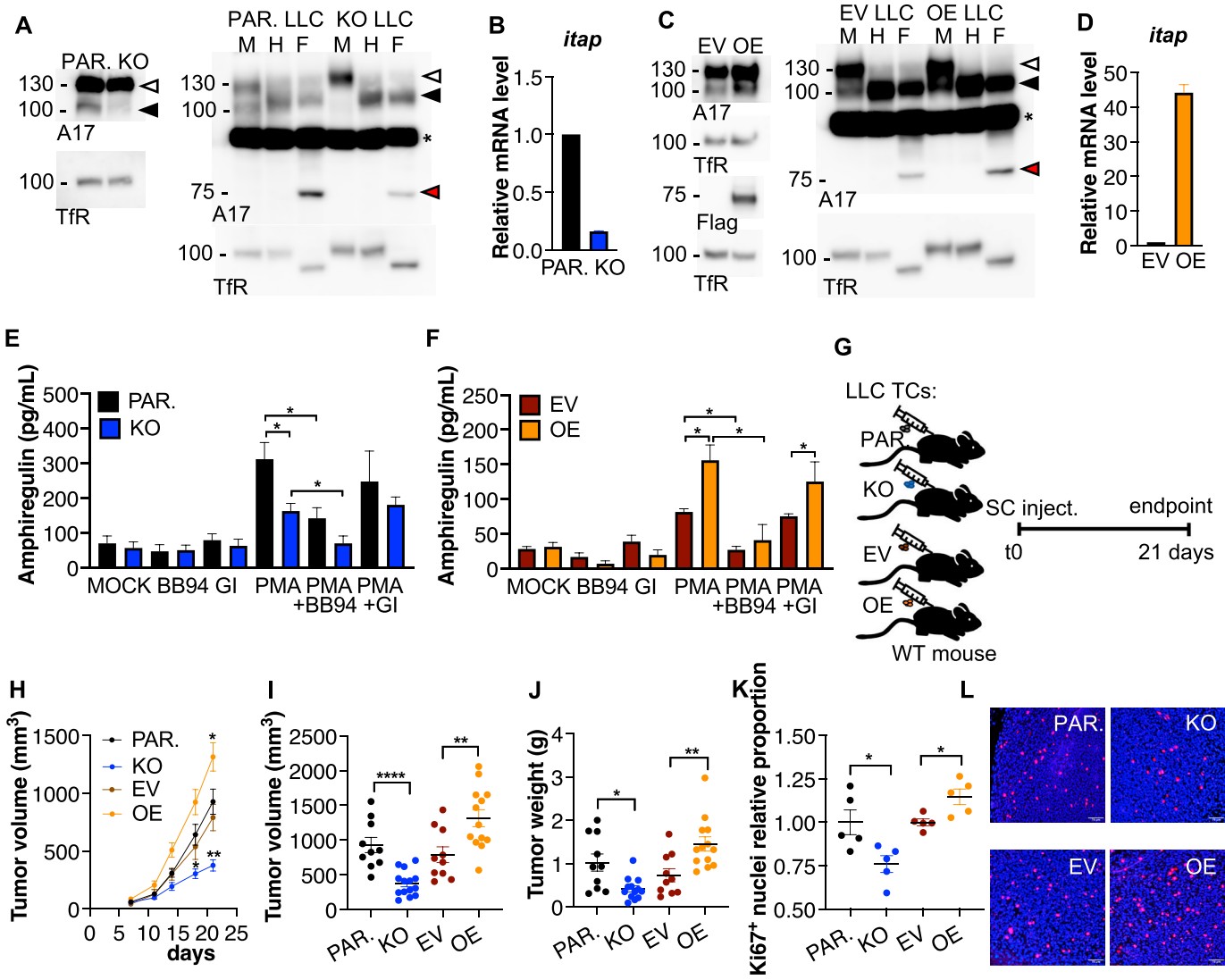

**Figure 5. Cell-autonomous expression of iTAP/Frmd8 promotes tumor growth.**
**(A, C)** Immunoblots showing expression levels of immature versus mature ADAM17 (A17) in parental (PAR.) versus iTAP/Frmd8 KO LLC cells (A) or empty vector (EV)–transduced versus iTAP/Frmd8-overexpressing (OE) LLC cells (C). Protein samples were deglycosylated with Endo-H (H) and PNGase F (F) versus mock (M). In (C), the left-hand immunoblot shows expression levels of Flag-tagged iTAP in EV-transduced versus iTAP/Frmd8-overexpressing (OE) LLC cells. Immunoblots for the transferrin receptor serve as a loading control. Immature ADAM17 is indicated with white arrowheads; the black arrowhead denotes both glycosylated mature ADAM17 and deglycosylated immature ADAM17, respectively (which have similar electrophoretic mobility), whereas red arrowheads denote the fully deglycosylated, mature, ADAM17 polypeptide. The asterisk indicates an unspecific band. **(B, D)** Relative mRNA expression levels of iTAP/Frmd8 in parental versus iTAP/Frmd8 KO LLC cells (B) and EV-transduced control cells versus iTAP/Frmd8-overexpressing (OE) LLC cells (D). Expression was normalized to the levels of a housekeeping control mRNA (*Gapdh*). **(E, F)** Soluble amphiregulin levels in the culture medium of parental versus iTAP/Frmd8 KO cells (E) or EV control cells versus iTAP/Frmd8-overexpressing (OE) LLC cells (F). The cells were preincubated with DMSO, the metalloprotease inhibitor MM (5 $\mu$M), or the ADAM10-specific inhibitor GI254023X (GI) (1 $\mu$M) and then treated with DMSO or PMA (1 $\mu$M). n = 3. **(G)** Schematic illustrating the protocol for the subcutaneous (SC) tumor model. **(H)** Volume of subcutaneous tumors in WT immunocompetent mice receiving iTAP/Frmd8 KO LLC cells or iTAP/Frmd8-overexpressing (OE) LLC cells over time compared with their respective controls. **(I, J)** Tumor volume (I) and tumor weight (J) of iTAP/Frmd8 KO LLC cells or iTAP/Frmd8-overexpressing (OE) LLC cells compared with their respective controls injected into WT mice. **(K, L)** Proportion of Ki67-positive nuclei relative to controls (K) and representative images (L) of the tumors described above. n = 3 with three to six mice per genotype per experiment. For Ki67 analysis, n = 2 with two to three mice per genotype. Results are indicated as the mean ± SEM. A scale bar is indicated within the Ki67 IF images of 78 $\mu$m. The Mann–Whitney–Wilcoxon test was used; * represents $P < 0.05$, ** represents $P < 0.01$, and **** represents $P < 0.0001$.

impairments and are defective in the shedding of TNF and L-selectin (Adrain et al, 2012; McIlwain et al, 2012; Siggs et al, 2012). Notably, ADAM17-deficient neutrophils exhibit alterations in rolling velocity and adhesion in response to inflammatory insults that can be normalized with L-selectin–neutralizing antibodies (Tang et al, 2011). In addition, there is a positive correlation between circulating soluble L-selectin levels and disease severity in human ulcerative colitis patients (Seidelin et al, 1998). Therefore, future studies on the role of iTAP/Frmd8 in inflammatory models, particularly those associated with leukocyte infiltration, are warranted.

Interestingly, iTAP/Frmd8 KO MEFs exhibited reduced levels of STING and blunted STING-associated interferon responses to HSV-1

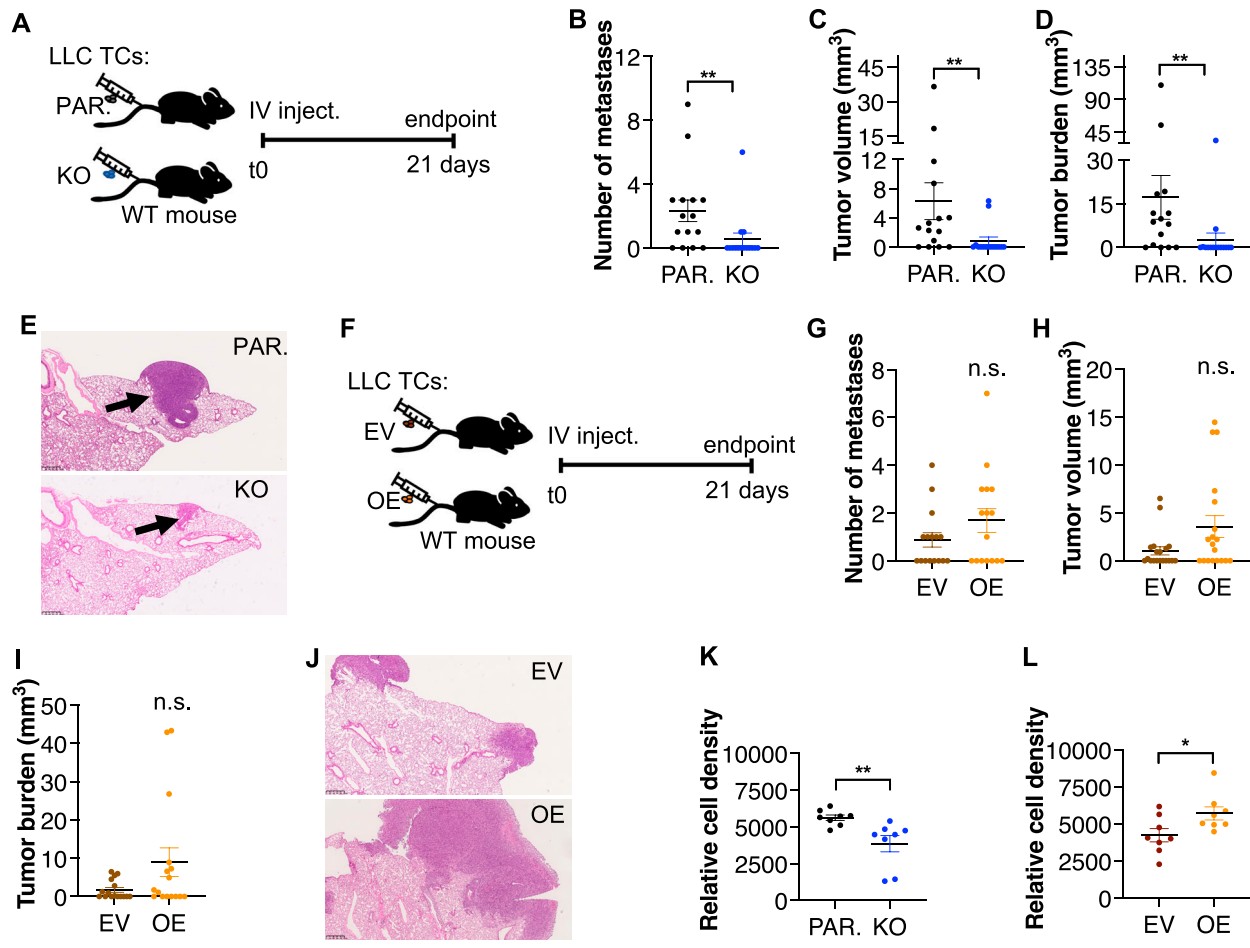

**Figure 6. iTAP/Frmd8 induces LL tumor cell proliferation and metastasis.**
**(A, F)** Schematic of the tumor model triggered by *i.v.* injection of parental (PAR.) versus iTAP/Frmd8 KO LLC cells into immunocompetent WT hosts (A) or *i.v.* injection of empty vector–transduced versus iTAP/Frmd8-overexpressing (OE) LLC cells into immunocompetent WT hosts (F). **(B, C, D, E, G, H, I, J)** Number of lung metastases (B, G), mean metastatic tumor volume (C, H), tumor burden (D, I), and H&E images (E, J) isolated from WT mice after tail vein injection with parental versus iTAP/Frmd8 KO (B, C, D, E) or empty vector–transduced versus iTAP/Frmd8-overexpressing (OE) (G, H, I, J) LLC cells. n = 3 with three to eight mice per group. **(K, L)** Relative cell density of parental versus iTAP/Frmd8 KO (K) or empty vector–transduced versus iTAP/Frmd8-overexpressing (OE) LLC cells (L) after 48 h of culture with medium supplemented with 0.5% FCS. Results are indicated as the mean ± SEM. A scale bar is indicated within the H&E images. The Mann–Whitney–Wilcoxon test was used; * represents *P* < 0.05, and ** represents *P* < 0.01.

infection (Fig 2) as has been reported for iRhom2-deficient cells (Luo et al, 2016). These data emphasize the importance of iTAP/Frmd8 as an integrator of two major iRhom2-dependent inflammatory pathways: TNF and STING, and emphasize the potential importance of iTAP/Frmd8 as an anti-inflammatory target.

Our observation that iTAP/Frmd8 KO mice exhibit a normal intestinal epithelium under control conditions but are sensitized to DSS-mediated colitis is consistent with the observation that mice homozygous for a hypomorphic mutation of ADAM17 (ADAM17$^{ex/ex}$) are defective in epithelial repair responses in response to DSS because of a failure to shed EGFR ligands (Chalaris et al, 2010). In agreement with this, the shedding of endogenous levels of the key EGFR ligand amphiregulin is defective in iTAP/Frmd8 KO LLC cells and increased in cells overexpressing iTAP/Frmd8 (Fig 5). Interestingly, unlike our observed sensitization of iTAP/Frmd8 KO mice to DSS-triggered colitis (Figs 3 and S5), it has been shown that iRhom2 KO mice are not sensitized to DSS-induced colitis (Siggs

et al, 2012; Geesala et al, 2019). A potential explanation for this discrepancy is that iRhom1, which plays a redundant role with iRhom2 in ADAM17 regulation, is expressed alongside iRhom2 in the intestinal tract, including the large intestine (Christova et al, 2013). iRhom1 expression may mask the impact of iRhom2 deletion, explaining the lack of sensitization of iRhom2 KO mice to DSS-induced colitis (Siggs et al, 2012; Geesala et al, 2019).

An obvious question concerns why on the one hand iTAP/Frmd8 KO mice are ostensibly normal unless challenged with inflammatory insults, yet on the other hand they exhibit substantially reduced levels of mature ADAM17 across a range of tissues. This includes tissues in which ADAM17 mutant mice themselves exhibit profound developmental phenotypes, including the heart, skin/keratinocytes (Fig S1), liver, and lung (Oikonomidi et al, 2018). One possibility is that the traces of mature ADAM17 that persist in iTAP/Frmd8 KO tissues are sufficient to prevent the pronounced embryonic phenotypes exhibited in ADAM17 KO or iRhom DKO models.

Indeed, ADAM17$^{ex/ex}$ mice, a hypomorphic mutant that expresses traces of ADAM17, are viable but exhibit eye, skin, and heart defects (Chalaris et al, 2010), indicating that mouse development is extremely sensitive to threshold levels of ADAM17 activity. Consistent with this notion, ADAM17$^{ex/ex}$ mice are more profoundly sensitized to DSS-induced colitis (Chalaris et al, 2010) than appears to be the case in iTAP/Frmd8 KO mice (Fig 3), suggesting that the titer of mature ADAM17 available may be higher in iTAP/Frmd8 KO mice than in ADAM17$^{ex/ex}$ mice. Potentially, the residual traces of mature ADAM17 that survive in iTAP/Frmd8 KO cells could be the result of redundancy between iTAP/Frmd8 and other FERM domain–containing molecules that may bind to and regulate the trafficking of the sheddase complex. Further studies will be required to clarify this. Given that the most pronounced impact of loss of iTAP/Frmd8 is in immune cells, it will be interesting to establish whether loss of iTAP/Frmd8 has a more pronounced impact on iRhom2-containing sheddase complexes compared with iRhom1-containing sheddase complexes. Although a preferential impact of iTAP/Frmd8 on iRhom2 biology is consistent with the lack of impact on mouse development and the inflammatory defects associated with iRhom2 KO mice, iTAP/Frmd8 binds to iRhom1 (Künzel et al, 2018; Oikonomidi et al, 2018) and ADAM17 maturation defects have been observed in lysates from the brains of iTAP/Frmd8 KO mice (Künzel et al, 2018), where iRhom1 is believed to play a more prominent role.

Interestingly, we note that there is strong co-expression between *iTAP/FRMD8* and iRhom1 (*RHBDF1*), iRhom2 (*RHBDF2*), or *ADAM17* genes in a range of major human tissues (e.g., pituitary, brain, spinal cord, stomach, kidney, heart, liver, lung, pancreas, and amygdala) (Table 1 and Fig S9A–C). Strikingly, in other tissues (e.g., visceral fat [omentum], salivary and mammary glands, tibial nerve, esophagus, intestine, female reproductive system, prostate, arteries, skin, and whole blood), there is much stronger co-expression correlation between *iTAP/FRMD8* and *RHBDF2* than between *iTAP/FRMD8* and *RHBDF1*. Indeed, in some cases (e.g., visceral adipose tissue, tibial artery, skin, vagina; Table 1 and Fig S9A) the co-expression coefficient between *iTAP/FRMD8* and *RHBDF1* is very weak or negative. There are also some (though fewer) instances where the expression between *iTAP/FRMD8* and *RHBDF1* correlates much better than it does with *RHBFD2*. As anticipated from previous functional studies (Christova et al, 2013; Li et al, 2015), this occurs in some (e.g., cerebellar hemisphere; Table 1 and Fig S9C) but not all anatomical regions in the brain. Relating these observations back to our present work, the higher and more frequent correlation between the expression of *iTAP/FRMD8* and *RHBDF2* and the correspondingly poorer co-expression with *RHBDF1* in some tissues could reconcile why most of the iTAP/Frmd8 KO phenotypes found in our study are immune-related. We also speculate that there may be tissues in which iTAP/Frmd8 is redundant with (an)other FERM domain–containing protein(s), whose expression may better correlate with iRhom1 than iTAP/Frmd8 does. Redundancy between iTAP/Frmd8 and similar molecules could explain why KO of iTAP/Frmd8 causes inflammatory defects but not ADAM17-associated epithelial developmental phenotypes.

An alternative explanation for the mild phenotypes observed in iTAP/Frmd8 KO mice could reflect the compartment within the cell where loss of iTAP/Frmd8 exerts its effects on the sheddase complex. Whereas KO (or DKO) of iRhom proteins results in a trafficking block early in the secretory pathway that prevents the generation of mature ADAM17 (Adrain & Freeman, 2012; McIlwain et al, 2012; Christova et al, 2013; Li et al, 2015), depletion of iTAP/Frmd8 triggers the precocious mis-sorting of the sheddase complex to lysosomes, wherein iRhom and ADAM17 are degraded (Künzel et al, 2018; Oikonomidi et al, 2018). As iTAP/Frmd8 appears to influence sheddase complex trafficking within the endocytic or endolysosomal pathway, iTAP/Frmd8's influence occurs after mature ADAM17 has been generated. Speculatively, this could allow small steady-state levels of ADAM17 substrate shedding to occur on or near the cell surface before the sheddase complex being mis-sorted to and degraded in lysosomes. This transient exposure of mature ADAM17 to its substrates could prevent the catastrophic loss of ADAM17 function phenotypes. Future studies will be required to define the precise trafficking route taken by the sheddase complex in normal cells and the defective itinerary incurred in iTAP/Frmd8 KOs.

Our study also establishes that iTAP/Frmd8 expression can influence multiple aspects of tumor growth and dissemination: it conditions the host microenvironment to promote tumor growth (Fig 4), while also influencing cell-autonomous tumor growth and dissemination (Figs 5 and 6). These observations are potentially consistent with ADAM17-associated defects in the tumor microenvironment in the cleavage of pro-inflammatory factors (e.g., TNF, IL-6R) (Schumacher & Rose-John, 2022) that can recruit inflammatory cells to drive tumor cell proliferation, or the release of growth factors such as amphiregulin (Fig 5E and F) to promote tumor cell growth.

Interestingly, mutations or copy-number variations within the *iTAP/FRMD8* gene have been reported in clones isolated from patients suffering from myeloid malignancies. This includes the preleukemic condition myelodysplastic syndrome, where *iTAP/FRMD8* mutations have been found in patient clones after chemotherapy (da Silva-Coelho et al, 2017). Moreover, amplifications in *iTAP/FRMD8* copy number have been observed in pediatric acute myeloid leukemia patients, particularly within the context of primary chemotherapy resistance (McNeer et al, 2019). Finally, in adult AML *iTAP/FRMD8* overexpression is associated with a poorer prognosis (Bou Samra et al, 2012). It will be interesting to determine, in future studies, whether regulation of the sheddase complex underpins the association between iTAP/Frmd8 and poor disease prognosis, or whether iTAP/Frmd8 regulates additional pathways within the context of myeloid malignancies.

One of the major challenges in the potential therapeutic modulation of ADAM17 activity concerns the poor preclinical performance (Murumkar et al, 2020; Calligaris et al, 2021) of numerous ADAM17-specific drugs, which have proved to be toxic, in part because of cross-reactivity with related metalloproteases. Indeed, given the crucial biological roles fulfilled by ADAM17 in multiple organs during development, it is arguably desirable to achieve partial—rather than complete—therapeutic ADAM17 inhibition or, indeed, to achieve tissue-specific blockade of ADAM17 activity. With this in mind, the targeting of iTAP/Frmd8 may be an appealing potential strategy to target ADAM17 activity in specific compartments during chronic inflammatory diseases or cancer, while

Table 1.  Correlation analysis (Pearson's coefficient) of *iTAP/FRMD8* with *iRHOM1* and *iRHOM2* and *ADAM17* gene expression in several human anatomical regions.

| | *iRHOM1* | | *iRHOM2* | | *ADAM17* | |
|---|---|---|---|---|---|---|
| Anatomical region | R | *P*-value | R | *P*-value | R | *P*-value |
| Visceral AT (Omentum) | −0.087 | 0.22 | 0.56 | 0 | 0.47 | $4.60 \times 10^{-12}$ |
| Salivary glands | 0.18 | 0.18 | 0.8 | $2.30 \times 10^{-13}$ | 0.87 | 0 |
| Mammary gland | 0.066 | 0.38 | 0.32 | $1.60 \times 10^{-5}$ | 0.28 | 0.00018 |
| Thyroid | 0.26 | $1.70 \times 10^{-5}$ | 0.46 | $4.40 \times 10^{-16}$ | 0.26 | $7.60 \times 10^{-6}$ |
| Adrenal gland | 0.26 | 0.0035 | 0.15 | $9.70 \times 10^{-2}$ | 0.21 | 0.016 |
| Pituitary | 0.65 | $3.20 \times 10^{-14}$ | 0.58 | $3.80 \times 10^{-11}$ | 0.71 | 0 |
| Hypothalamus | 0.56 | $5.30 \times 10^{-8}$ | 0.51 | $9.80 \times 10^{-7}$ | 0.32 | 0.0032 |
| Cerebellar hemisphere | 0.62 | $1.10 \times 10^{-11}$ | 0.14 | $1.60 \times 10^{-1}$ | 0.29 | 0.0038 |
| Frontal cortex | 0.54 | $6.30 \times 10^{-9}$ | 0.5 | $6.90 \times 10^{-8}$ | 0.47 | $7.00 \times 10^{-7}$ |
| Brain caudate nucleus | 0.67 | $2.20 \times 10^{-15}$ | 0.8 | 0 | 0.77 | 0 |
| Putamen | 0.62 | $4.50 \times 10^{-10}$ | 0.76 | 0 | 0.58 | $1.80 \times 10^{-8}$ |
| Substantia nigra | 0.61 | $5.30 \times 10^{-7}$ | 0.64 | $7.70 \times 10^{-8}$ | 0.72 | $3.00 \times 10^{-10}$ |
| Hippocampus | 0.61 | $6.50 \times 10^{-10}$ | 0.63 | $1.90 \times 10^{-10}$ | 0.68 | $8.30 \times 10^{-13}$ |
| Tibial nerve | 0.16 | 0.009 | 0.48 | 0 | 0.28 | $1.70 \times 10^{-6}$ |
| Spinal cord | 0.58 | $1.20 \times 10^{-6}$ | 0.53 | $1.20 \times 10^{-5}$ | 0.62 | $1.40 \times 10^{-7}$ |
| Esophagus | 0.11 | 0.18 | 0.73 | 0 | 0.52 | $6.60 \times 10^{-11}$ |
| Stomach | 0.58 | 0 | 0.62 | 0 | 0.77 | 0 |
| Ileum | −0.37 | $2.50 \times 10^{-4}$ | 0.28 | $6.40 \times 10^{-3}$ | 0.18 | 0.092 |
| Sigmoid colon | 0.12 | 0.16 | 0.56 | $3.10 \times 10^{-13}$ | 0.52 | $4.20 \times 10^{-11}$ |
| Ovary | 0.082 | 0.44 | 0.46 | $5.80 \times 10^{-6}$ | 0.31 | 0.0038 |
| Fallopian tubes | 0.032 | 0.96 | 0.7 | $1.90 \times 10^{-1}$ | −0.74 | 0.15 |
| Uterus | 0.37 | $9.00 \times 10^{-4}$ | 0.14 | $2.30 \times 10^{-1}$ | 0.15 | 0.18 |
| Cervix | −0.79 | 0.061 | 0.91 | $1.20 \times 10^{-2}$ | 0.95 | 0.004 |
| Vagina | −0.086 | 0.43 | 0.59 | $2.70 \times 10^{-9}$ | 0.46 | $8.60 \times 10^{-6}$ |
| Testes | 0.04 | 0.61 | 0.16 | $4.50 \times 10^{-2}$ | 0.37 | $1.10 \times 10^{-6}$ |
| Prostate | 0.29 | 0.0039 | 0.7 | $4.40 \times 10^{-16}$ | 0.65 | $2.10 \times 10^{-13}$ |
| Bladder | 0.55 | 0.13 | 0.91 | $7.20 \times 10^{-4}$ | 0.89 | 0.0013 |
| Kidney cortex | 0.87 | $2.00 \times 10^{-9}$ | 0.79 | $5.10 \times 10^{-7}$ | 0.85 | $7.00 \times 10^{-9}$ |
| Tibial artery | −0.084 | 0.16 | 0.7 | 0 | 0.49 | 0 |
| Aorta | 0.13 | 0.054 | 0.59 | 0 | 0.33 | $1.40 \times 10^{-6}$ |
| Coronary artery | 0.048 | 0.61 | 0.44 | $5.70 \times 10^{-7}$ | 0.32 | 0.00048 |
| Heart atrial appendage | 0.62 | 0 | 0.7 | 0 | 0.72 | 0 |
| Heart left ventricle | 0.44 | $6.50 \times 10^{-11}$ | 0.61 | 0 | 0.67 | 0 |
| Liver | 0.57 | $7.80 \times 10^{-11}$ | 0.73 | 0 | 0.71 | 0 |
| Skin (no sun) | 0.073 | 0.27 | 0.39 | $6.70 \times 10^{-10}$ | 0.42 | $1.30 \times 10^{-11}$ |
| Lung | 0.65 | 0 | 0.51 | 0 | 0.35 | $7.00 \times 10^{-10}$ |
| Pancreas | 0.54 | 0 | 0.51 | 0 | 0.64 | 0 |
| Skeletal muscle | 0.24 | $1.60 \times 10^{-6}$ | 0.27 | $3.30 \times 10^{-8}$ | 0.37 | $2.50 \times 10^{-14}$ |
| Spleen | 0.36 | 0.00021 | 0.37 | 0.00013 | 0.38 | $8.70 \times 10^{-5}$ |
| Amygdala | 0.82 | 0 | 0.8 | $2.20 \times 10^{-16}$ | 0.79 | $8.90 \times 10^{-16}$ |
| Whole blood | 0.23 | $2.40 \times 10^{-5}$ | 0.64 | 0 | 0.85 | 0 |

avoiding collateral impact on the vital functions of ADAM17 in normal tissues.

# Materials and Methods

### Experimental animals

iTAP/Frmd8 KO mice were generated as previously described on a C57BL/6J background (Oikonomidi et al, 2018). Mice were maintained in a SPF facility on a 12-h light/dark cycle, at standard subthermoneutral conditions of 20–24°C and an average of 50% humidity, in ventilated cages with corn cob as bedding. We co-housed C57BL/6J WT and iTAP/Frmd8 KO animals together to homogenize differences in microbiota and other environmental conditions. Food intake and body weight were recorded weekly from the age of 5 wk and for 9–10 wk.

### Mouse tumor model

Two cancer models were used, based on the inoculation of murine LLC (ATCC# CLR1642) cells. To assess topical growth, 0.5 million LLC cells in PBS were inoculated subcutaneously (100 $\mu$l) into the flank of WT versus iTAP/Frmd8 KO mice. Tumor dimensions were measured with a caliper 1 wk after inoculation, and afterward twice a week until 21 d after, when the mice were euthanized and the tumor and lungs were collected. Tumor weight, number, volume, and burden were evaluated, as was the presence of metastasis in the lungs as previously described (Mendonça et al, 2019). To assess LLC cell migration to the lung, in a second model, 0.5 million LLC cells in PBS were inoculated intravenously into the tail vein (200 $\mu$l), and after 21 d, the mice were euthanized, after which the lungs were collected and their tumors counted and measured. For both models, tumor volume was calculated using formula V = 0.52 × a × b$^2$, where a and b equal the longer and shorter length of the tumor, respectively.

### Acute colitis model

To induce intestinal inflammation, 8- to 12-wk-old iTAP/Frmd8 KO and WT mice were given 3% DSS in the drinking water for a total of 8 d. The DSS solution was replenished every 2 d. Body weight was measured daily. At day 8, water was given to the mice, and at day 10, the mice were euthanized and the large intestine was collected, and its length was measured as described (Wirtz et al, 2017). For recovery assessment, the mice mentioned above were given 2% DSS in the drinking water for 5 d, and afterward, the DSS was replaced by water. The mice were euthanized at day 10. Body weight was measured three times per week, and the colon length was measured.

### Sepsis model

LPS was injected intraperitoneally into 8- to 12-wk-old iTAP/Frmd8 KO and WT mice at a dosage of 37.5 mg/Kg in PBS. Serum was collected via cardiac puncture from animals that only received PBS (0), and from mice administrated with LPS at 3 and 6 h post-LPS injection. TNF and IL-6 levels were measured from serum using specific ELISA kits (88-7324-22 and 88-7064-88, respectively; eBioscience).

### Western blotting

Cells isolated from iTAP/Frmd8 KO and WT mice were washed in PBS and lysed for 10 min on ice in TX-100 lysis buffer (1% Triton X-100, 150 mM NaCl, and 50 mM Tris–HCl, pH 7.4) containing protease inhibitors and 100 $\mu$M of the metalloprotease inhibitor 1,10-phenanthroline. 1,10-Phenanthroline was added because without it ADAM17 autocatalytically cleaves off its cytoplasmic tail, resulting in the loss of the epitope detected by the anti-ADAM17 antibody (Ab39162; Abcam) (Adrain et al, 2012). The same was applied to LLC cells and MEFs. Tissues from iTAP/Frmd8 KO and WT mice were lysed in a modified RIPA buffer (1% Triton X-100, 150 mM NaCl, 50 mM Tris–HCl, pH 7.4, 1 mM EDTA, 1% sodium deoxycholate, and 0.1% SDS) containing protease inhibitors and 0.1 mM of 1,10-phenanthroline using a TissueLyser II (QIAGEN). The samples were then quantified and normalized, and to improve the detection of ADAM17, glycoproteins were captured using 40 $\mu$l of concanavalin A (ConA) agarose. Beads were washed twice in the same buffer supplemented with 1 mM EDTA, 1 mM MnCl$_2$, and 1 mM CaCl$_2$ and eluted by heating for 15 min at 65°C in a sample buffer supplemented with 15% sucrose. After centrifugation at 1,000$g$ for 2 min, the supernatants were collected. In the same cases, the denatured lysates were digested for 2 h at 37°C with deglycosylating enzyme endoglycosidase H (Endo-H), which removes high mannose N-linked glycans added in the ER, but not complex N-linked glycans found in the later secretory pathway, and with PNGase F, which deglycosylates both. Then, the samples were denatured for 5 min at 65°C. The lysates were fractionated by SDS–PAGE and transferred onto PVDF membranes. After blocking with 5% milk in TBS-T for 30 min, the membranes were cut and incubated overnight with the following primary antibodies: rabbit anti-ADAM17 (1:1,000, Ab39162; Abcam), mouse anti-Flag HRP (1:1,000, A8592; Sigma-Aldrich), rat anti-tubulin (1:1,000, Clone YL1/2, IGC antibody facility), mouse anti-transferrin receptor (1:1,000, 13-6800; Life Technologies), rabbit anti-STING (D2P2F) (1:1,000, 13647; Cell Signaling Technology), rabbit anti-ATPase 5A (1:500, Ab151229; Abcam), mouse anti-p97 ATPase (1:1,000, 65278; Progen), mouse anti-EMC3 (1:100, Sc-365903; Santa Cruz Biotechnology), mouse anti-TOMM20 (1:1,000, WH0009804M1; Sigma-Aldrich), rabbit anti-$\beta$-actin (1:2,500, Ab8227; Abcam), rabbit anti-TMEM97 (1:600,

Level of correlation of *iTAP/FRMD8* with *iRHOMs* and *ADAM17* gene expression in a range of human body tissues. Pearson's correlation analysis using data from the Genotype-Tissue Expression project on the Gene Expression Profiling Interactive Analysis tool demonstrating that there is a strong co-expression between *iTAP/FRMD8* and iRhom1 (*RHBDF1*), iRhom2 (*RHBDF2*), or *ADAM17* genes in numerous human anatomical regions. AT indicates adipose tissue.

26444-1-AP; Proteintech), rabbit anti-ATP13A1 (1:700, 16244-1-AP; Proteintech), rabbit anti-TMBIM6 (1:1,000, 26782-1-AP; Proteintech), rabbit anti-moesin (1:1,000, 3150; Cell Signaling Technology), rabbit anti-UBAC2 (aa250-372) (1:500, produced by Peter J Espenshade's Lab) (Lloyd et al, 2013), rabbit anti-GAPDH (1:2,500, 2118; Cell Signaling Technology). The next day, the membranes were washed and incubated with the secondary antibody anti-rabbit HRP (1:5,000, 1677074P2; Cell Signaling Technology), anti-mouse HRP (1:5,000, 1677076P2; Cell Signaling Technology), or anti-rat HRP (1:5,000, IGC antibody facility). After washing, protein bands were detected using ECL. Semi-quantitative densitometric analysis on images from Western blot exposures was performed with Fiji software.

### Histopathological analysis

Histopathological analysis was performed by a veterinary pathologist in a blinded manner. Sections were examined under a Leica DM LB2 microscope. Collected samples were formalin-fixed, paraffin-embedded, sectioned (3-$\mu$m sections), and stained with H&E (Sigma-Aldrich). LLC tumors and dorsal skin were embedded in 3.5–4% agar. Each specimen was sectioned into a minimum of eight slices of equal thickness. Depending on the size of the sample, they were sectioned into 1.3- or 2.6-mm slices. Whole slide images were acquired with a Hamamatsu Nanozoomer slide scanner. Measurements were performed with Visiopharm stereology software. Volume was estimated according to the Cavalieri principle, in which a grid of equally spaced points is overlaid onto systematic fields of view, automatically generated by the software. Each point has an associated area. Every time a point hits the tissue of interest, it is counted. The volume of the tissue of interest equals the number of points ($Qi$) multiplied by the area per point ($a/p$), multiplied by the section thickness. $V = Qi \times a/p \times T$. Lungs were exhaustively sectioned, and every 30[th] section, a slice was chosen, resulting in a total of around 20 sections per lung, separated by 90 $\mu$m. A grid of 5 × 5 points was used for the tumors, and that of 3 × 3 points was used to evaluate the lungs. In the case of the murine LLC allograft model, every time a point hits necrosis or superficial ulceration, it was counted separately. The left-hand top corner point was used for tumor estimation (the area of this point is 25 times larger than the rest of the points). The measurements were done across 20–30% of the section's region of interest in fields of view with 10× magnification for the tumors.

The large intestines from the colitis model were scored as described (Seamons et al, 2013). Specifically, the proximal, mid-, and distal colon was scored by the level of mucosal loss (0, none; 1, mild [<5%]; 2, moderate [6–30%]; 3, marked [31–60%]; and 4, severe [>61%]), hyperplasia (0, none; 1, mild; 2, moderate; 3, severe with aberrant crypts; and 4, severe with herniation), inflammation (0, none; 1, mild mucosa only; 2, moderate mucosa and submucosa; 3, severe with abscesses and erosion; and 4, severe with obliteration of architecture and ulceration or transmural), and extent of intestine affected and of the intestine affected by the most severe score (0, none; 1, <5%; 2, 6–30%; 3, 31–60%; and 4, >60%). The IBD total score resulted from the sum of the scores for the proximal, mid-, and distal colon.

The WT and iTAP/Frmd8 KO dorsal skin was sectioned at 250 $\mu$m, resulting in a total of five to eight slabs per skin. The slabs were then paraffin-embedded and photographed for volume estimation by point counting as described above. Eight paraffin sections (5 $\mu$m) were sampled from the upper-cut surface of each slab and stained with H&E. Estimations of follicle number were performed using the dissector principle, where the distance between the reference and the look-up section must represent 30% of the object's height. In this case, optimal distance was found to be in the range of 30–40 $\mu$m. If a particle is seen in only one of the sections, it is counted. The number of differences between the upper and lower sections is given by Q. $\hat{N} = \frac{\sum Q}{n x V(dis)}$, where V(dis) is the reference volume, and n is the number of sections. The follicle number was obtained by multiplying the numerical density by the total skin volume. Estimates of skin and epidermal volume were obtained according to the Cavalieri principle (Gundersen & Jensen, 1987). $V = T \times a/p \times \sum Pi$, where T is the thickness of the slabs, Pi is the number of points hitting the structures of interest, and ($a/p$) is the area associated with each point. After paraffin embedding and because of expected tissue shrinkage, the thickness was remeasured. The epidermal fraction was estimated by dividing the points hitting the epidermis by the points hitting the skin. $V = Pi(Epidermis)/Pi(Skin)$. To estimate surface area, a cycloidal test grid was superposed on sections. A reference volume was obtained by the Cavalieri principle. Surface density was then estimated by counting the number of intersections (I) between the cycloids and the epidermal/dermal interface. $\hat{S} = \frac{2 \times I}{\frac{1}{p} x \sum Pi}$, where I is the number of intersections counted, Pi is the number of points counted, and l/p is the test line length per point at the level of the tissue. The surface area was obtained by multiplying the surface density by the total skin volume. To facilitate interpretation, all previous measurements were converted into structure of interest per mm³. The stereological measurements were done with the software STEPanizer.

### Immunofluorescence

Tumors samples were deparaffinized and hydrated in PBS, and then in PBS with 0.1% of Triton X and 0.1% of Tween-20. Antigen retrieval was performed with citrate buffer for 50 min at 99°C in a water bath. Cooled samples were washed with PBS twice and exposed to blocking reagent (PBS with 0.1% of Triton X, 0.1% of Tween-20, 5% BSA, and 1:100 Fc Block, clone 2.4G2 produced in-house), for 1 h. After two washes in PBS, the samples were incubated overnight with the primary antibody rabbit anti-Ki67 (1:200, Ab16667; Abcam) diluted in PBS with 0.1% of Triton X and 0.1% of Tween-20 at 4°C. The next day, after three washes in PBS, the samples were incubated with a secondary goat anti-rabbit A594 antibody (1:350, A11012; Thermo Fisher Scientific) for 1 h at room temperature. After three washes in PBS, the samples were incubated with DAPI (0.15 $\mu$g/ml; Invitrogen) for 10 min. After two washes, the samples were mounted with Vectashield Antifade Mounting Medium (H100-10; VectorLabs). Representative and aleatoric images were taken at 200× magnification under the microscope Zeiss Imager2/Apotome2, and the number of

positive nuclei for Ki67 per image was analyzed with QuPath (Bankhead et al, 2017). A macro was used to count the nuclei, which can be provided upon request.

## HSV-1 infection of mouse embryonic fibroblasts

iTAP/Frmd8 KO, iRhom2 KO, and WT MEFs (generated as previously described) (Christova et al, 2013; Oikonomidi et al, 2018) were cultured in DMEM high glucose (Biowest), supplemented with 10% FCS (Biowest) and 1% Pen/Strep (Sigma-Aldrich). Cells were plated in a density of 650,000 cells in a six-well plate. The next day, the cells were infected with HSV-1 as described (Colgrove et al, 2016) at an MOI of 3 or maintained in their medium as a control. After an hour of infection, the medium was changed to normal maintenance medium, and 4 h post-infection, the medium was collected and cleared to measure the level of IFN-$\beta$ using an ELISA kit (DY8234-05; R&D Systems). RNA was extracted from the same cells 4 h post-infection using a kit from NZYTech.

## Generation of iTAP/Frmd8 KO and overexpressing (OE) LLC cell lines

Murine LLC cells were cultured in DMEM high glucose (Biowest), supplemented with 10% FCS (Biowest) and 1% Pen/Strep (Sigma-Aldrich).

To generate iTAP/Frmd8 KO LLC cell line, sgRNAs targeting coding exon 3 (5′-CCAGACCTTACCCAAGAGAG-3′, 5′-CGATGTCCTGGTGTACCTGG-3′) and exon 4 (5′-CATGGCGACATCATCATCGG-3′) were cloned into a plasmid px330 (Zhang Lab, 42230; Addgene). HEK293FT cells ($6 \times 10^6$) were transfected with a pMD-VSVG envelope plasmid, psPAX2 helper plasmid, and pLentiCas9-Blast plasmid (52962; Addgene) or pLentiGuide-puro (52963; Addgene) plus the 3 iTAP/Frmd8 sgRNA sequences inserted into CRISPR plasmid px330 using Fugene HD-based lipofection. After 48 h, the virus was collected. Ultracentrifuged, 150× concentrated lentivirus LentiCas9-Blast (20 $\mu$l) was added to 180,000 LLC cells supplemented with 8 $\mu$g/ml polybrene. After selection with blasticidin (16 $\mu$g/ml), the cells were left for 1 wk to allow Cas9 expression. Then, lentivirus LentiGuide-puro plus the three iTAP/Frmd8 sgRNAs were added (2 ml, in two rounds with a 24-h interval) to 180,000 LLC cells expressing LentiCas9 supplemented with polybrene. The cells were then selected with puromycin (6 $\mu$g/ml) and plated in a limiting dilution in 96-well plates to isolate single clones. The clones were screened by qRT-PCR to confirm depletion of iTAP/Frmd8 mRNA levels. Three clones with the strongest iTAP/Frmd8 deletion were combined into a pool, and the parental Cas9-expressing cells were used as controls.

To generate a LLC cell line overexpressing iTAP/Frmd8, HEK293 ET cells ($6 \times 10^6$) were transfected with the plasmids pCL-Eco packaging plasmid (Naviaux et al, 1996) plus pM6P.BLAST empty vector (kind gift of F Randow), or pM6P containing the mouse iTAP/Frmd8 WT cDNA fused to a Flag tag using Fugene HD-based lipofection. After 48 h, the retroviruses were collected and added (2 ml, twice sequentially, with a 24-h interval) to 180,000 LLC cells supplemented with polybrene 8 $\mu$g/ml. The cells were then selected with puromycin (6 $\mu$g/ml). The impact upon ADAM17 maturation and *iTAP/Frmd8* mRNA levels were assessed by Western blotting and

qRT-PCR, respectively. These cell lines were then used in mouse experiments.

LLC iTAP/Frmd8 KO and OE versus their respective controls (PAR—parental; and EV—empty vector) were plated in a 24-well plate at a density of 700,000. After 4 h, the cells were starved in serum-free medium overnight and then incubated for 1 h with the ADAM17 inhibitor batimastat (BB-94) at 10 $\mu$M (196440; Calbiochem), with the selective ADAM10 inhibitor GI254023X (GI) at 1 $\mu$M (SML0789; Sigma-Aldrich) or DMSO (67-68-5; Sigma-Aldrich). Then, the cells were stimulated with PMA at 1 $\mu$M (P1585; Sigma-Aldrich) or DMSO for 1 h. The medium was collected and, after clearance, was used to evaluate the levels of TNF, HB-EGF, and amphiregulin using specific ELISA kits (88-7324-22, eBioscience; DY8239-05, R&D Systems; and DY989, R&D Systems, respectively).

## Isolation and differentiation of primary cells in vitro

BMDMs and BMDCs were isolated from 8- to 10-wk-old iTAP/Frmd8 KO and WT mouse tibias and femurs, and BMDMs were differentiated as previously described (Adrain et al, 2012) in RPMI 1640 with GlutaMAX (Biowest), supplemented with 10% FCS (Biowest), 1% Pen/Strep and gentamicin sulfate (10 $\mu$g/ml) (Sigma-Aldrich), 20% of L929 cell (CCL-1; ATCC) conditioned medium, and 50 $\mu$M 2-mercaptoethanol (Sigma-Aldrich). At day 7 of differentiation, 0.5 million cells were plated in a 24-well plate. At day 8, cells were stimulated with LPS at 0.25, 0.5, and 1 $\mu$g/ml, and the supernatant was collected after 2 and 6 h. TNF and IL-6 were measured from clarified supernatants using specific ELISA kits (88-7324-22 and 88-7064-88, respectively; eBioscience). BMDCs were differentiated for 8 d in RPMI 1640 with GlutaMAX (Biowest), supplemented with 10% FCS (Biowest), 1% Pen/Strep (Sigma-Aldrich), 20 ng/ml GM-CSF, and 50 $\mu$M 2-mercaptoethanol (Sigma-Aldrich). Keratinocytes were isolated from 1- to 2-d-old iTAP/Frmd8 KO and WT neonatal epidermis following the protocol of CELLnTEC. Briefly, the isolated skin was incubated overnight at 4°C in a tube containing 10 ml of 1× Dispase (Roche) and 2× Pen/Strep (Sigma-Aldrich) and 2× Amphotericin B (Life Tech). The next day, the skin was transferred to CnT-07 medium (CELLnTEC) and the dermis was separated. Accutase (CnT-Accutase-100; CELLnTEC) was then added to the epidermis for a period of 20–30 min at room temperature. CnT-07 medium was then added, and the epidermis was separated into single cells. After washing and resuspension, the cells were cultured in CnT-07 medium supplemented with IsoBoost (CnT-ISO-50; CELLnTEC) for 3 d.

## Flow cytometry

Spleens and mesenteric, inguinal, cervical, and axillary lymph nodes were mashed through a 70-$\mu$m cell strainer. The digested cells were centrifuged (5 min, 4°C, 400$g$), and red blood lysis buffer was added (0.88% NH$_4$Cl). The cells were washed and filtered through a 70-$\mu$m strainer. Three million splenocytes were plated per condition in RPMI medium (Biowest) supplemented with 2% FCS (Biowest). Samples were incubated for 1 h with the ADAM17 inhibitor marimastat at 5 $\mu$M (444,289-5; Calbiochem, Merck Millipore), with the selective ADAM10 inhibitor GI254023X (GI) at 1 $\mu$M (SML0789; Sigma-Aldrich) or DMSO (67-68-5; Sigma-Aldrich). Afterward, the cells were stimulated with PMA

at 1 $\mu$M (P1585; Sigma-Aldrich) or LPS (0.25 $\mu$g/ml) or DMSO for 2 h. Samples were centrifuged and resuspended in PBS supplemented with 2% FCS and incubated in Fc Block, clone 2.4G2 (1:100, produced in-house), for 15 min. After washing, staining was performed by incubation for 30 min with anti-mouse CD45.2-PE, clone 104.2; B220-PE-Cy7, clone RA3-6B2; L-selectin–7-AAD, clone MEL-14; CD4-FITC, clone GK1.5; CD8-Pacific Blue; clone YTS169.4; and CD3-A647, clone 500A2 (all at 1:100; produced in-house) and with the Zombie Aqua fixable dye (423101; BioLegend).

For B-cell sorting, the total amount of splenocytes and lymphocytes collected was washed and incubated in Fc Block for 15 min. Then, the samples were washed and incubated for 30 min with anti-mouse CD45.2-PE; clone 104.2; CD19-PE-Cy7, clone 6D5; and CD3-A647, clone 500A2 (all at 1:100; produced in-house) and with the Zombie Aqua fixable dye (423101; BioLegend).

The purity of DCs was assessed by staining for anti-mouse CD45.2-A647, clone 104.2; CD11b-FITC, clone M1/70; and CD11c-PE, clone HL3 (all at 1:100; produced in-house).

LLC iTAP/Frmd8 KO and OE versus their respective controls (PAR. and EV) were plated in six-well plates at a density of 0.5 million cells. After 4 h, the cells were starved in serum-free medium overnight and afterward maintained in medium containing 0.5% FCS for 48 h. To compare the relative cell density per well, cells were trypsinized, washed, and stained with DAPI (0.2 $\mu$g/ml; Invitrogen) after which 25 $\mu$l of Precision Count Beads (BioLegend) was added to each culture suspension as an internal standard for cell counting by flow cytometry. All the samples were analyzed using a Fortessa X20 with FlowJo software, version 10.2, to determine the relative cell density in each well.

### Quantitative transcriptional analysis

LLC tumors and lungs from iTAP/Frmd8 KO and WT mice were snap-frozen in liquid nitrogen and kept at –80°C until RNA extraction (NZYTech). The same was applied to LLC cells and MEFs. First-strand cDNA was synthesized from total RNA using the SuperScript III First-Strand Synthesis SuperMix. Real-time PCR analysis was performed using the comparative $C_T$ method (Schmittgen & Livak, 2008). Gene expression was normalized to *Gapdh* mRNA levels. Primer sequences are listed in Table 2.

### Correlation analysis

Pearson's correlation analysis of *iTAP/FRMD8* versus *RHBDF1*, *RHBDF2*, or *ADAM17* mRNA expression in several human anatomical regions was performed using data from the Genotype-Tissue Expression project (GTEx Consortium, 2013) on the Gene Expression Profiling Interactive Analysis tool (Tang et al, 2019) available at http://gepia2.cancer-pku.cn/#correlation

### Statistics

To compare single measurements between control and test groups, the Mann–Whitney–Wilcoxon test was used. For repeated measurements, the two-way ANOVA was used. The statistical analysis was performed using GraphPad Prism, version 6. Results are presented as the average ± SEM. *P*-values

**Table 2.** List of primers used in the quantitative transcriptional analysis.

| List of primers | |
|---|---|
| **Oligonucleotides** | **Sequence** |
| *Gapdh* forward | 5'-AACTTTGGCATTGTGGAAGG-3' |
| *Gapdh* reverse | 5'-ACACATTGGGGGTAGGAACA-3' |
| *Il1β* forward | 5'-GAAGAAGAGCCCATCCTCTG-3' |
| *Il1β* reverse | 5'-TCATCTCGGAGCCTGTAGTG-3' |
| *Il6* forward | 5'-ACGGCCTTCCCTACTTCACA-3' |
| *Il6* reverse | 5'-CATTTCCACGATTTCCCAGA-3' |
| *Il10* forward | 5'-GCTCTTACTGACTGGCATGAG-3' |
| *Il10* reverse | 5'-CGCAGCTCTAG GAGCATGTG-3' |
| *Mcp1* forward | 5'-GGAAAAATGGATCCACACCTTGC-3' |
| *Mcp1* reverse | 5'-TCTCTTCCTCCACCATGCAG-3' |
| *Cxcl2* forward | 5'-CAGGGGCTGTTGTGGCCAGTG-3' |
| *Cxcl2* reverse | 5'-CCCAGGCTCCTCCTTTCCAGGT-3' |
| *Ccl4* forward | 5'-TCGTGGCTGCCTTCTGTGCTC-3' |
| *Ccl4* reverse | 5'-CTGAAGTGGCTCCTCCTGCCC-3' |
| *Frmd8* forward | 5'-AGGCAACAACAGTGAACGTG-3' |
| *Frmd8* reverse | 5'-GTCAACCTCGCCATGGAA-3' |
| *Tnfa* forward | 5'-ATGAGCACAGA AAGCATGATC-3' |
| *Tnfa* reverse | 5'-TACAGGCTTGTCACTCGAATT-3' |
| *Pcna* forward | 5'-CGAAGCACCAAATCAAGAGA-3' |
| *Pcna* reverse | 5'-CCCGACGGCATCTTTATTAC-3' |
| *Il6* forward | 5'-ACGGCCTTCCCTACTTCACA-3' |
| *Il6* reverse | 5'-CATTTCCACGATTTCCCAGA-3' |
| *Ifnb1* forward | 5'-CAGCTCCAAGAAAGGACGAAC-3' |
| *Ifnb1* reverse | 5'-GGCAGTGTAACTCTTCTGCAT-3' |
| *UL-42* forward | 5'-TGTTCACCACGAGTACCTGC-3' |
| *UL-42* reverse | 5'-TTTCCCCGTACACCGTCTTG-3' |
| *ICP22* forward | 5'-GAAATCTCCGATGCCACCGA-3' |
| *ICP22* reverse | 5'-TCTGGGGTTTCCAGCGTAAC-3' |

Sequence of primers used for the gene expression analysis on the following samples: LLC tumors and lungs from iTAP/Frmd8 KO and WT mice; parental versus iTAP/Frmd8 KO LLC cells and empty vector–transduced versus iTAP/Frmd8-overexpressing (OE) LLC cells; and iTAP/Frmd8 KO and iRhom2 KO versus WT MEFs.

< 0.05 were represented as (*), <0.01 as (**), <0.001 as (***), and <0.0001 as (****).

### Study approval

Animal procedures were approved by the National Regulatory Agency (DGAV—Direção Geral de Alimentação e Veteriná́ria) and by the Ethics Committee of Instituto Gulbenkian de Ciência and the Institutional Animal Care, and were carried out in accordance with the Portuguese (Decreto-Lei no. 113/2013) and European (Directive 2010/63/EU) legislation related to housing, husbandry, and animal welfare.

# Supplementary Information

# Acknowledgements

The authors thank the Animal Facility, the Histopathology, the Advanced Imaging, and Flow cytometry units and the antibody service of the Instituto Gulbenkian de Ciência. We thank Cristina Branco for reagents and Sarah Maguire for advice about expression correlation analysis. We thank Colin Crump (Cambridge, UK) for providing the HSV-1 DNA. We thank Stefan Rose-John and Cristina Branco for advice concerning the LLC cell model and helpful discussions. We acknowledge the support of Fundação Calouste Gulbenkian; Queen's University Belfast; Worldwide Cancer Research (14–1289); a Marie Curie Career Integration Grant (project no. 618769); and Fundação para a Ciência e Tecnologica (FCT) (grants SFRH/BCC/52507/2014, PTDC/BEX-BCM/3015/2014, and LISBOA-01–0145-FEDER-031330), and funding from "La Caixa" Foundation under the agreement (LCF/PR/HR17/52150018). This work was developed with the support of the research infrastructure Congento, project LISBOA-01–0145-FEDER-022170, co-financed by Lisboa Regional Operational Programme (Lisboa 2020), under the Portugal 2020 Partnership Agreement, through the European Regional Development Fund (ERDF) and Foundation for Science and Technology (Portugal).

## Author Contributions

M Badenes: conceptualization, data curation, formal analysis, validation, investigation, visualization, methodology, project administration, and writing—original draft, review, and editing.
E Burbridge: conceptualization, data curation, formal analysis, investigation, methodology, and writing—review and editing.
I Oikonomidi: conceptualization, data curation, formal analysis, investigation, methodology, and writing—review and editing.
A Amin: conceptualization, data curation, formal analysis, investigation, methodology, and writing—review and editing.
E de Carvalho: investigation and methodology.
L Kosack: investigation.
C Mariano: investigation and methodology.
P Domingos: conceptualization and methodology.
P Faísca: resources, data curation, formal analysis, investigation, and methodology.
C Adrain: conceptualization, resources, formal analysis, supervision, funding acquisition, investigation, methodology, project administration, and writing—original draft, review, and editing.

## Conflict of Interest Statement

The authors declare that they have no conflict of interest.

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
