## [Reviewer comments · Life Science Alliance]

Life Science Alliance

The ADAM17 sheddase complex regulator iTAP/Frmd8 modulates inflammation and tumor growth

Marina Badenes, Emma Burbridge, Ioanna Oikonomidi, Abdulbasit Amin, Érika de Carvalho, Lindsay Kosack, Camila Mariano, Pedro Domingos, Pedro Faisca, and Colin Adrain

DOI: <https://doi.org/10.26508/lsa.202201644>

Corresponding author(s): Colin Adrain, Gulbenkian

Review Timeline:

Submission Date:	2022-08-02
Editorial Decision:	2022-08-04
Revision Received:	2022-12-14
Editorial Decision:	2022-12-15
Revision Received:	2022-12-22
Accepted:	2023-01-03

Transaction Report:

Please note that the manuscript was reviewed at Review Commons and these reports were taken into account in the decision-making process at Life Science Alliance.

General Statements

Dear Editor,

We are pleased to submit an article entitled '**The ADAM17 sheddase complex regulator iTAP/Frmd8 modulates inflammation and tumor growth**' for consideration. Below I describe the background to our work, its key findings, importance for several fields and its suitability for *Life Science Alliance*.

Background. ADAM17 (also called TACE) is a cell surface protease that “sheds”, i.e., proteolytically releases, a range of important signaling molecules. ADAM17 controls several key biological pathways that are important for development, immune responses, inflammation and epithelial repair. Prominent ADAM17 substrates include the inflammatory cytokine TNF (Tumor Necrosis Factor) and its receptors (TNFRI, TNFRII); the IL6 receptor, the activating ligands (e.g. Amphiregulin, TGF α , HB-EGF) of the EGFR (Epidermal Growth Factor Receptor) and the cell adhesion molecule L-Selectin. Excessive shedding of ADAM17 substrates plays a key role in chronic inflammatory diseases and cancer. Work from several laboratories, including ours, established that ADAM17 functions within the context of a molecular assemblage called the “Sheddase Complex”. Importantly, integral membrane proteins called iRhoms form the essential core of the Sheddase Complex, without which ADAM17 trafficking and activity is abolished.

We recently discovered a new interactor and regulator of the sheddase complex called iTAP/Frmd8 (Oikonomidi et al., eLIFE, 2018). Our work established that iTAP/Frmd8 is required to maintain the cell surface stability of the Sheddase Complex, preventing the precocious shunting of ADAM17 and iRhom2 to lysosomes and their consequent degradation therein. Ablation of iTAP/Frmd8 results in a loss of ADAM17 activity. As the pathophysiological role(s) of iTAP/Frmd8 have so far been barely addressed, our present MS characterizes the impact of loss of iTAP/Frmd8 *in vivo*, focusing on its impact on the known pathophysiological roles of ADAM17.

Key findings and significance of our work. Our MS describes an extensive characterization of iTAP/Frmd8-null mice that were generated by our laboratory (Oikonomidi et al., eLIFE, 2018). In brief, our work demonstrates that iTAP/Frmd8 acts as an important endogenous regulator of ADAM17 in several key contexts *in vivo*. Loss of iTAP/Frmd8 prevents the shedding of native ADAM17 substrates (e.g. L-selectin, Tumor Necrosis Factor) from immune cells. We also find that similar to iRhom2 deletion, iTAP/Frmd8 ablation decreases the amount of STING protein in mouse embryonically fibroblasts, identifying it as an important regulator of this innate immune pathway. In addition, iTAP/Frmd8 deletion substantially inhibits the transcription of interferon-inducible genes (Il-6 and Ifnb1) in response to Herpes virus simplex type 1 (HSV-1) infection. Hence our data establish iTAP/Frmd8 as a key regulator of STING responses to DNA viruses. We also observe that loss of iTAP/Frmd8 renders mice more prone to a model of experimental

colitis, as has been observed in ADAM17 mutant mouse studies. Significantly, we show that iTAP/Frmd8 is required for the secretion of the endogenous EGFR ligand Amphiregulin, indicating that iTAP/Frmd8 regulates EGFR signaling. Finally, we observe that iTAP expression influences tumor growth *in vivo*, in a cell-autonomous manner, as well as by influencing the tumor microenvironment.

The major implications of our work are:

- 1) Our data establish that iTAP/Frmd8 is a major regulator of the Sheddase Complex function *in vivo* that controls ADAM17 biology in key contexts including inflammation, and tumor growth/dissemination.
- 2) Notably, unlike ADAM17-deficient mice, iTAP KO mice develop normally, suggesting that iTAP plays a specialized role in ADAM17 regulation in specific key contexts/tissues *in vivo*.
- 3) ADAM17, which is implicated in inflammatory disease and cancer, has been the subject of extensive attempts by pharma to develop safe and specific ADAM17 inhibitors. However, a major drawback associated with targeting ADAM17 is the toxicity of multiple ADAM17 inhibitory compounds. This is associated with the cross-reactivity of inhibitory compounds with other metalloproteases (e.g. matrix metalloproteases). Our work suggests that pharmacological intervention at the level of iTAP may be beneficial to target ADAM17 activity in specific tissues/compartments during chronic inflammatory diseases or cancer, avoiding the deleterious impact on vital functions associated with the widespread inhibition of ADAM17 in normal tissues.

Summary of key observations.

- iTAP/Frmd8 deletion impairs ADAM17 maturation and L-selectin shedding in immune cells.
- iTAP/Frmd8 regulates the STING pathway, and inhibits transcription of the immunomodulatory genes *Il-6* and *lfnb1* triggered by HSV-1 infection, as has been reported for *iRhom2*.
- iTAP/Frmd8 KO immune cells (including macrophages) cannot secrete TNF upon exposure to LPS, *in vitro* and *in vivo*. This establishes iTAP is a regulator of inflammatory responses *in vivo*.
- iTAP/Frmd8 deletion exacerbates DSS-induced colitis in mice, as has been reported for ADAM17-mutant mice.
- iTAP/Frmd8 expression promotes ADAM17-mediated secretion of the EGFR ligand Amphiregulin in tumor cells.
- Deletion of iTAP/Frmd8 in the tumor microenvironment inhibits tumor growth.
- Depletion of iTAP/Frmd8 in tumor cells inhibits tumor cell proliferation *in vitro* and tumor growth and dissemination *in vivo*.

Revision Plan

Our work has broad significance for the life science, across several fields, since it establishes the pathophysiological importance of iTAP/Frmd8 in the biological processes mediated by ADAM17, a protease that as noted above, controls inflammatory and growth-promoting pathways in multiple contexts. Our work highlights the importance of iTAP/Frmd8 as an important regulator of ADAM17 across several contexts and scales, from cells to the whole organism. Our manuscript reveals new insights into the iTAP regulation of ADAM17 sheddase complex modulation of pathophysiological pathways—a field that we have contributed to significantly in the past (Adrain et al., *Science*, 2012; Christova, Adrain et al., *EMBO. Rep* 2013; Cavadas et al., *Cell Rep.* 2017; Oikonomidi et al., *eLIFE*, 2018; Badenes et al., *Molec. Metabolism*, 2020) and which continues to attract a lot of interest.

In the attached documents you will find an unmarked copy of the MS, as well as a copy in which all changes made to the initial version are marked in red text.

Description of the planned revisions

Responses to Reviewer 1

Reviewer 1/Point 1: “Is Frmd8/iTAP always co-expressed with iRhom2 and ADAM17?”

Response: We thank the reviewer for this interesting question, which was partly addressed in our previous work (Figure S1B of Oikonomidi et al., *eLIFE*, 2018). To bolster this information, we propose to interrogate public gene expression databases to establish the co-expression patterns of mouse and human Frmd8/iTAP, ADAM17, iRhom1 and iRhom2.

Comment. *In Fig. 2E-J, the authors employ a DSS-driven inflammatory bowel disease model. It has been shown before (Chalaris et al, 2010; cited in the manuscript) that the higher susceptibility of hypomorphic ADAM17 mice was related to reduced shedding of EGF-R ligands in this model. Therefore, the authors should address shedding of these ligands in Frmd8/iTAP knock-out mice.*

Response. We agree that it would be interesting to determine whether the enhanced susceptibility is related to reduced shedding of EGFR ligands. Chalaris et al. (Figure 4A, 2010, *JEM*) showed that the overall levels of the EGFR ligand TGF α was elevated in the colons of ADAM17 ex/ex mice when exposed to DSS. We will carry out a similar IHC staining for TGF α in our model, as well as imaging Ki67 as a marker for cellular proliferation associated with epithelial

Revision Plan

repair. This will complement our data (Fig. 5E,F) that show the defective shedding of the endogenous EGFR ligand AREG from mouse Lewis lung carcinoma cells.

Comment. *In the experiment shown in Fig. 5, the authors inject parental and Frmd8/iTAP knock-out LLC tumor cells into WT mice. The note that the number of metastases, tumor volume and tumor burden is dramatically decreased. In the study by Bolik et al, 2022 (cited in the manuscript) it has been shown that in hypomorphic ADAM17 mice, metastasis formation by LLC tumor cells was dramatically reduced. In this study it was also shown that ADAM17 activity in endothelial cells was responsible for this effect, which was at least in part mediated by TNF-RI and TNF-RII. This mechanistic difference should be addressed in the manuscript.*

Comment. *Along the same line: when tumor cells are injected i.v., the cells need to extravasate before they can form tumors. The authors need to mechanistically address whether the effects of Frmd8/iTAP are on extravasation or on tumor growth (or both).*

Response. The reviewer addresses an interesting mechanistic question: whether iTAP expression in endothelial cells modulates tumor extravasation and mechanistically whether this depends on the TNF signaling pathway. We argue that it this would be better mechanistically addressed in cell culture models similar to those used by Bolik et al. (2021, JEM). We will assess the transwell migration of tumor cells through monolayers of endothelial cells in which Frmd8/iTAP expression is ablated. The dependency on the TNF-associated endothelial cell necroptosis will also be addressed. In addition, the experiments mentioned in the response to the point made by reviewer No. 2 (below) has now addressed tumor proliferation more thoroughly.

Responses to Reviewer 2

Response: We have now addressed all comments of Reviewers 2

Responses to Reviewer 3

7. Fig. 4C: the increase in the 75 kDa fragment upon iTAP OE is difficult to see. Can you quantify the increase? And also the reduction in the KO cells? **Response:** we will incorporate this quantification into the revised MS.

Description of the revisions that have already been incorporated in the transferred manuscript

Responses to Reviewer 1

Comment. *It has been shown that iRhoms have additional clients apart from ADAM17. For instance, the adaptor protein STING has been reported to be constitutively associated with iRhom2. Therefore, it is possible that Frmd8/iTAP also plays a role in the STING pathway. This point needs to be addressed.*

Revision Plan

Response. The referee raises an interesting and relevant point. We have addressed this by testing whether iTAP/Frdm8 KO cells are defective in the induction of interferon responses in response to Herpes simplex virus type 1 (HSV-1), a DNA virus that engages STING responses. New Figure 2 now shows that, similar to iRhom2-deficient cells, iTAP/Frdm8 deficiency reduces the steady state levels of STING and impairs STING-associated responses (e.g. upregulation of IL6 and IFNbeta mRNA levels in response to HSV-1 infection). We thank the reviewer for this suggestion and believe that these new data will further bolster the pathophysiological importance of iTAP/Frdm8 as an iRhom regulator and more broadly, as a regulator of two key innate immune pathways.

Comment. All Western blots shown in the figures and supplemental figures should be quantified by a suitable software such as Image J.

Response. We thank the reviewer for the suggestion. These have been incorporated into the current submitted version.

Comment. In Fig. 2C,D, the authors use a sepsis model and they show that Frmd8/iTAP knock-out mice have lower TNFa levels than WT mice. Is this also true for sIL-6R levels?

Response. We have currently shown that the shedding of endogenous TNF, L-Selectin and AREG is impaired in iTAP/Frdm8 KO. The reviewer would like to know whether cleavage of another ADAM17 substrate, the IL-6R, is reduced in our model. We have carried out this experiment *in vivo* and find that the loss of iTAP/Frdm8 does not influence the shedding of IL6R under steady-state conditions, nor in response to I.P. injection of LPS. This figure (below) has not been added to the main MS.

iTAP/FRMD8 KO and WT mice have similar serum levels of IL6R under steady state conditions and after exposure to LPS

Revision Plan

IL-6R levels in the serum of WT versus iTAP/Frmd8 KO mice following *i.p.* injection of PBS (0) or LPS (37,5 µg/g) for 3 or 6h. n= 2 with 3-5 mice per genotype per condition. Results are indicated as mean ± SEM. Mann-Whitney-Wilcoxon test was used. sIL-6R levels were quantitated using an ELISA kit (Mouse IL-6R alpha DuoSet ELISA DY1830, R&D).

Reviewer 1 Minor points:

1. The authors name the protein Frmd8/iTAP sometimes as Frmd8 and sometimes as iTAP. This is confusing for the reader. Since the protein has been characterized under both names, the authors should stick to Frmd8/iTAP. **Response:** now corrected throughout, to iTAP/Frmd8.
2. Along the same line: the authors should stick to the name ADAM17 and not sometimes switch to the older name TACE. **Response:** now corrected throughout.
3. The authors use Frmd8/iTAP knock-out mice. It is not clear from the statement of p5, whether they use the mice described in Künzel et al, 2018 or the mice described in Oikonomidi et al, 2018. This should be clarified. **Response:** now corrected-we used the mice generated by our group, described in Oikonomidi et al.
4. Some references (e.g. Dong et al, 1999 and Gschwind et al, 2003) are incomplete. **Response:** now corrected throughout

Responses to Reviewer 2

Point 1. Figure 1: The findings regarding L-selectin shedding are very clear and perhaps meaningful. However, a discussion putting these findings in context with the disease models used later in the manuscript is warranted. Currently, inclusion of these data does not add much to the story if not discussed or referenced later in the manuscript.

Response. We agree that these data are striking and that they were not properly placed within context in the paper. These data have now been discussed more extensively in the Results and Discussion sections of the revised version of the paper.

Reviewer 2. Point 2 Figure 2 [now Figure 3]: The authors observe that iTAP KO mice have worse outcomes following DSS-colitis. In the text, they mention that iRhom2 KO mice do not phenocopy the iTAP KO mice following DSS-colitis, yet no explanation is offered. If the mechanism of iTAP is proposed to be through iRhom2 activity and ADAM17 shedding, you would expect the iRhom2 KOs to demonstrate similar intestinal phenotypes. The authors should comment on this discrepancy.

Response: We apologize for not proposing a potential explanation for this in the original version. Our published data (Christova et al., 2013, EMBO Rep.) show that iRhom1 is expressed alongside iRhom2 in the large intestine; iRhom1 levels may be sufficient to prevent ADAM17-associated defects in iRhom2 KO mice in response to experimental colitis. We have adjusted the narrative in the results section (page 9) to note this potential explanation.

Comment. Figure 3: The authors make the statement that "...although inflammatory infiltrates were modest in the lungs of mice..." Is this based on histology alone? If the authors want to make this claim, they must assess immune infiltrates directly (e.g. using flow

Revision Plan

cytometry). **Response:** This is a fair point—it was based on histology alone. We have now removed this sentence.

Comment. *The authors evaluate lung metastasis in the LLC subcutaneous model but any conclusion about metastasis cannot be made in this model without looking at primary tumors of a similar size. Metastasis tends to be associated with the size of the primary tumor so smaller primary tumors usually mean lower levels of metastasis (without being able to parse apart direct effects on the metastatic process). I assume that the data in Fig 3H are from mice with different tumor sizes--in order to properly evaluate this, the authors need to euthanize WT and KO animals with similar tumor burdens and compare metastatic burden.*

Response: We thank the reviewer for pointing this out. We agree and have now removed the data in original Fig. 3H and the text associated with this.

Comment. *Similar to Figure 3, claims about metastasis cannot be made from these experiments without comparing mice with similar primary tumor burden. The metastasis data in Figure 5 are much more solid and convincing.* **Response:** We agree and have removed the relevant data (Fig. 3H,I and Supplementary Fig. 5) and the associated text.

Comment. *Figure 4: Claims about proliferation cannot be made here because the results as shown are not significant (IFig 4K [now current Fig. 5K]). Additional readouts for proliferation should be used to support this conclusion.*

Response: We agree and have now added IF images showing Ki67 expression (Fig. 5K,L) plus data showing the mRNA levels of the proliferative marker PCNA (Fig. S7C).

Comment. *Including total mRNA levels of cytokines does not add to this figure. First, bulk levels of mRNA are not a good way to evaluate the state of a tumor (immune cell phenotype/activity would be better). Second, TNF and IL-6 were used in previous figures as readouts of ADAM17 activity (or not) and here are just markers of inflammation? This is confusing/contradictory. If included, this should be moved to the supplement.*

Response: Yes, the mRNA levels of TNF and IL6 were used as general readouts for inflammation here. We agree that, in our context, this could be confusing, while relying elsewhere on shedding of the TNF protein as a readout for ADAM17 activity. We have moved these mRNA-based data into supplementary Fig. 5A,B to minimize confusion.

Minor Comments.

The language regarding any results that are not statistically-significant need to be softened in the text. In several places, there are statements about non-significant results that are much too definitive and somewhat misleading. Non statistically-significant results can be useful to include to show trends (as in Fig 5 G-I), but the interpretation should not be overstated.

Response: Our apologies that we did not get the balance right. We hope that we have now toned the relevant comments down.

Revision Plan

Comment. *Figure 5: The authors use Fig 5 K & L as evidence that tumor cells proliferated more or less rapidly, depending on expression levels of iTAP. The data do not support this statement. If I understand the methods correctly, this assay involves plating of 500K tumor cells and then harvesting after several days. Upon harvest, there were 100-fold fewer cells (~5K). To me this indicates effects on survival, not proliferation. Proliferation was never measured in this assay. Without these data, the authors can make no claim regarding the mechanisms of tumor cell autonomous functions of iTAP. explain better the experiment.*

Response: We apologize for the ambiguity in how we described this experiment. The assay involved plating 500,000 tumor cells per well. After approximately 72h, the cells were trypsinized and an aliquot of each cell suspension was supplemented with a defined volume of counting beads. The cell density in each sample, normalized against a defined number of counting beads (500) was determined by FACS. Hence, the readout is the relative cell density between individual wells and is not, as aberrantly stated an indication of the absolute number of cells in the wells. We have now clarified this in the relevant figure, figure legends and materials and methods.

The title is overstated. In this manuscript, the authors do not show clear mechanistic links for iTAP promoting epithelial repair (worse outcomes after DSS are not just caused by decreased repair). The strongest data in the manuscript are those regarding tumor growth. This should be highlighted in the title

Response: Thanks for the suggestion. We have now modified the title, to remove references to epithelial repair and to emphasize the impact on tumor growth.

Responses to Reviewer 3 (minor points)

Comment. *My only major concern is the choice of the control LLC cells. The cells used are not the ideal control for the iTAP knock-out cells. Both wild-type and iTAP knock-out cells were transduced with a Cas9 expressing vector, as it should be. But only the knock-out cells were further transduced with a virus expressing the gRNAs against iTAP, whereas the control cells apparently were not transduced with a virus expressing control gRNAs. Three single cell clones of the iTAP ko cells were pooled for in vivo injection, whereas the parental pool (and not clones) where apparently used as a control. My concern is that the ko cells do not only differ from the wild-type cells due to the knock-out of iTAP, but potentially also due to other gene expression alterations resulting from the additional transduction of the knock-out cells with a gRNA virus and because of the selection of single cell clones. Such expression changes beyond the simple lack of iTAP may have a major influence on those tumor phenotypes in vivo, where these cells were used. Ideally, the authors would generate an additional, independent pool of iTAP knock-out cells and repeat one of the crucial in vivo experiments. As a time-saving alternative, the authors need to demonstrate that the iTAP knock-out cells are nearly identical to the control cells (with the exception of iTAP). This could be done by RNA sequencing or cell lysate proteomics or by blotting for several different proteins (at least 10 from different compartments) and demonstrating that there is no significant change in protein abundance - apart from iTAP.*

Revision Plan

Response: While the *in vivo* experiments used cas9-expressing cells as the control, the equivalent *in vitro* experiments used the cas9-expressing cells further transduced to express the empty vector, which we agree is a better control. As we observe analogous results compared to their respective controls in the *in vivo* setting compared to the *in vitro* setting, we are confident that these results are genuine. Moreover, our *in vivo* and *in vitro* experiments that compare vector versus Frmd8/iTAP-overexpressing cells show the anticipated opposite effect compared to the KO cells, bolstering our confidence in these observations. Nonetheless, as suggested, we now provide western blot images and their associated densitometric analysis showing that the expression levels of 13 different proteins from a range of compartments are unaltered (Fig S6 A-M).

1. Indicate the concentrations of the used drugs (marimastat, PMA) in the figure legends.

Response: Apologies for this omission, which we have now corrected

2. Indicate in the manuscript that LLC cells are of mouse origin. **Response:** Apologies for this omission, which was now incorporated

3. Page 6, top paragraph: it is not clear to me, whether there is an eye phenotype or not. Please rephrase this sentence. **Response:** we have corrected this to make it clearer that the iTAP KO eyes develop normally (as per WT animals).

4. Figure legend 1: "...with 3 replicates per experiment". Indicate whether this refers to biological or technical replicates. **Response:** we have corrected this-they were technical replicates.

5. Indicate in figure legends which statistical test was used. **Response:** we have now indicated this throughout.

6. Fig. 2F [current figure 3F]. The y-axis label should be body weight and not body weight loss. **Response:** now corrected.

Description of analyses that authors prefer not to carry out

Please include a point-by-point response explaining why some of the requested data or additional analyses might not be necessary or cannot be provided within the scope of a revision. This can be due to time or resource limitations or in case of disagreement about the necessity of such additional data given the scope of the study. Please leave empty if not applicable.

Responses to Reviewer 1

Reviewer 1/Point 1. Although the Frmd8/iTAP protein was identified as a binding partner of the iRhom2/ADAM17 complex, it remains unclear whether this protein also serves as a binding partner of other proteins. When analyzing Frmd8/iTAP knock-out mice, this might be an important aspect, which is not addressed in the manuscript.

Response: We agree that this might be an important aspect and indeed have engaged collaborations with colleagues to do this in future. However, in our experience (e.g. Cavadas et al., Cell Reports, 2017; Oikonomidi et al., eLIFE 2018) experiments designed to objectively

Revision Plan

identify, then triage and ultimately validate interaction partners are a long-term project. Such experiments could take in excess of a year to complete from beginning to end. While it may be possible to obtain a list of untested putative interactors in a shorter timeframe, in the absence of rigorous validation of the hits, this would be a superficial contribution that risks clouding the literature rather than clarifying whether Frmd8/iTAP has additional physiologically relevant binding partners. Hence, we argue that such studies, although definitely pertinent, fall outside of the scope of our current study.

Reviewer 1/Point 2. *Was survival of the mice affected by the absence of Frmd8/iTAP?*

Response: Although this is an interesting question, our animal license does not permit us to use death as an endpoint in sepsis assays. Therefore, for animal ethics reasons, this experiment is unfortunately beyond our remit.

Comment. *Additionally, the conclusion about the importance of iTAP in intestinal repair would be better supported if the DSS colitis experiments were continued to later time points to include the recovery phase (once the mice return to original body weight), rather than just ending the experiment at peak repair.*

Response: Unfortunately, our animal license does not permit additional permutations in the protocol (e.g. extending beyond the timepoints post-DSS that we currently use). Nonetheless, as this experimental modification is predicted not to surpass moderate severity limits, we have queried with our institute's ethics committee whether permission could be sought to do this. We await the response. However, we argue that the current experimental setup includes a component of recovery, since the animals have been allowed to continue for 2 days post-DSS exposure. We believe that the experiments that will assess whether TGF α levels accumulate in tissue sections from DSS-treated mice, as well as the assessment of whether iTAP KO colonic epithelia undergo reduced proliferation, will help to strengthen the premise that the colonic phenotype is associated with defective EGFR-associated epithelial repair.

Responses to Reviewer 2. *We aim to address all of the revisions suggested by the reviewer-see above.*

Responses to Reviewer 3. *We aim to address all of the revisions suggested by the reviewer-see above.*

August 4, 2022

Re: Life Science Alliance manuscript #LSA-2022-01644

Dr. Colin Adrain
Instituto Gulbenkian de Ciencia
Rua da Quinta Grande 6
Oeiras 2780-156
Portugal

Dear Dr. Adrain,

Thank you for submitting your manuscript entitled "The ADAM17 sheddase complex regulator iTAP/Frmd8 modulates inflammation, epithelial repair, and tumor growth" to Life Science Alliance. We invite you to re-submit the manuscript, revised according to your Revision Plan.

Thank you for this interesting contribution to Life Science Alliance. We are looking forward to receiving your revised manuscript.

Sincerely,

B. MANUSCRIPT ORGANIZATION AND FORMATTING:

Reviewer #1 notes that “This is an interesting study, which addresses the important role of ADAM17 and the pathways controlled by this protease.” We thank the reviewer for their insightful critiques, which we believe have helped us to improve our MS substantially.

Major points:

Comment 1. *Although the Frmd8/iTAP protein was identified as a binding partner of the iRhom2/ADAM17 complex, it remains unclear whether this protein also serves as a binding partner of other proteins. When analyzing Frmd8/iTAP knock-out mice, this might be an important aspect, which is not addressed in the manuscript.* **Response:** We agree that this may be an important aspect and indeed have engaged collaborations with colleagues to do this in future. However, in our experience (e.g. Cavadas *et al.*, 2017; Oikonomidi *et al.*, 2018) experiments designed to objectively identify, then triage and ultimately validate interaction partners are a long-term project. Such experiments could take in excess of a year to complete from beginning to end. While it may be possible to obtain a list of untested putative interactors in a shorter timeframe, in the absence of rigorous validation of the hits, this would be a superficial contribution that risks clouding the literature rather than clarifying whether Frmd8/iTAP has additional physiologically relevant binding partners. Hence, we argue that such studies, although definitely pertinent, fall outside of the scope of our current study.

...Is Frmd8/iTAP always co-expressed with iRhom2 and ADAM17?

Response: We thank the reviewer for raising this interesting and very pertinent point. We have now addressed this by examining the co-expression of iTAP/FRMD8 with iRHOM1 (RHBDF1), iRHOM2 (RHBDF2) and ADAM17 respectively at the mRNA level in a panel of human tissues using the GEPIA2 algorithm (available at <http://gepia2.cancer-pku.cn/#correlation>). This analysis has now been incorporated into **Supplementary Fig. S9, Table 2** and in the Discussion. In brief, we observe several key points:

- 1) There is generally a strong positive correlation between the expression of iTAP/FRMD8 and iRHOM1 (RHBDF1), or iRHOM2 (RHBDF2) and ADAM17 in the majority of tissues tested.
- 2) The correlation in expression between iTAP/FRMD8 and RHBDF2 is often better than between iTAP/FRMD8 and RHBDF1. Indeed, there are several examples where the correlation of expression between iTAP/FRMD8 and RHBDF1 is poor (or negative) while the correlation between FRMD8 and RHBDF2 is strongly positive. Our speculative interpretation is that there may be tissues/organs (e.g. the skin, **Supplementary Fig. S9A and Table 2**) with redundancy between iTAP/FRMD8 and another FERM domain-containing protein, whose expression may better correlate with RHBDF1 than iTAP/FRMD8 does. This putative redundancy between iTAP/FRMD8 and similar molecules may explain why ablation of iTAP/Frmd8 causes inflammatory defects but not ADAM17-associated epithelial developmental phenotypes.
- 3) In a minority of cases, the correlation between iTAP/FRMD8 expression and RHBDF1 expression is greater than that of iTAP/FRMD8-RHBDF2 co-expression. This implies an inverse scenario to that discussed above in (2), e.g. that in a small number of tissues an iTAP/Frmd8 homolog may be co-expressed with and regulate iRhom2.

We have added the following new text:

“Interestingly, we note that there is strong co-expression between iTAP/FRMD8 with iRhom1 (RHBDF1), iRhom2 (RHBDF2) or ADAM17 genes in a range of major human tissues (e.g. pituitary, brain, spinal cord, stomach, kidney, heart, liver, lung, pancreas, and amygdala (Table 2, Supplementary Fig. 9A). Strikingly, in other tissues (e.g. visceral fat (omentum), salivary glands, mammary gland, tibial nerve, esophagus, intestine, female reproductive system, prostate, arteries, skin and whole blood), there is much stronger co-expression correlation between iTAP/FRMD8 and RHBDF2 than between iTAP/FRMD8 and RHBDF1. Indeed, in some cases (e.g. visceral adipose

tissue, tibial artery, skin, vagina; Table 2, Supplementary Fig. 9A) the co-expression coefficient between iTAP/FRMD8 and RHBDF1 is very weak or negative. There are also some (though fewer) instances where the expression between iTAP/FRMD8 and RHBDF1 correlates much better than it does with RHBDF2. As anticipated from previous functional studies (Christova et al., 2013; Li et al., 2015), this occurs in some (e.g. cerebellar hemisphere, Table 2, Supplementary Fig. 9C) but not all anatomical regions in the brain. Relating these observations back to our present work, the higher and more frequent correlation between expression of iTAP/FRMD8 and RHBDF2 and the correspondingly poorer co-expression with RHBDF1 in some tissues could reconcile why most of the iTAP/Frmd8 KO phenotypes found in our study are immune related. We also speculate that there may be tissues in which iTAP/Frmd8 is redundant with (an)other FERM domain-containing protein(s), whose expression may better correlate with iRhom1 than iTAP/Frmd8 does. Redundancy between iTAP/Frmd8 and similar molecules could explain why KO of iTAP/Frmd8 causes inflammatory defects but not ADAM17-associated epithelial developmental phenotypes.”

Comment 2. It has been shown that iRhoms have additional clients apart from ADAM17. For instance, the adaptor protein STING has been reported to be constitutively associated with iRhom2. Therefore, it is possible that Frmd8/iTAP also plays a role in the STING pathway. This point needs to be addressed.

Response. The referee raises a very interesting and relevant point. We have addressed this by testing whether iTAP/Frmd8 KO cells are defective in the induction of interferon responses in response to Herpes simplex virus type 1 (HSV-1), a DNA virus that engages STING responses. New Figure 2 now shows that, similar to iRhom2-deficient cells, iTAP/Frmd8 deletion reduces the steady state levels of STING and impairs STING-associated responses (e.g. upregulation of IL6 and IFN β mRNA levels in response to HSV-1 infection). We thank the reviewer for this suggestion. These new data will further highlight the pathophysiological importance of iTAP/Frmd8 as an iRhom regulator and more broadly, as a regulator of two key innate immune pathways (e.g. release of TNF and mobilization of STING-mediated IFN responses).

Comment 3. All Western blots shown in the figures and supplemental figures should be quantified by a suitable software such as Image J. **Response.** We thank the reviewer for the suggestion. These have been incorporated into the revised version.

Comment 4. In Fig. 2C,D, the authors use a sepsis model and they show that Frmd8/iTAP knock-out mice have lower TNF α levels than WT mice. Is this also true for sIL-6R levels? **Response.** We have currently shown that the shedding of endogenous TNF, L-selectin and AREG is impaired in iTAP/Frmd8 KOs. The reviewer would like to know whether cleavage of another ADAM17 substrate, the IL-6R, is reduced in our model. This is an interesting point and we have carried out this experiment in an *in vivo* model driven by LPS administration. We find that KO of iTAP/Frmd8 does not influence the shedding of IL6R under steady-state conditions, nor in response to I.P. injection of LPS (Fig. R1, below). This Figure has not been added to the MS. We currently do not understand the basis for the apparent specificity of the impact of loss of iTAP/Frmd8 on some, but not all ADAM17 substrates, but argue that this falls outside of the scope of the current MS.

Fig R1. iTAP/Frmd8 KO and WT mice have similar serum levels of IL6R under steady state conditions and after exposure to LPS

WT versus iTAP/Frmd8 KO mice were injected *i.p.* with PBS (0) or with LPS (37,5 μ g/g of body weight) for 3 or 6h. sIL-6R levels in the sera were quantitated using an ELISA kit (Mouse IL-6R alpha DuoSet ELISA DY1830, R&D). n= 2 experiments, 3-5 mice per genotype per condition. Results are indicated as mean \pm SEM. Mann-Whitney-Wilcoxon test was used.

Was survival of the mice affected by the absence of Frmd8/iTAP? **Response:** Although this is an interesting question, our animal license does not permit us to use death as an endpoint in sepsis assays. Therefore, for animal ethics reasons, this experiment is unfortunately beyond our remit.

5. In Fig. 2E-J, the authors employ a DSS-driven inflammatory bowel disease model. It has been shown before (Chalaris et al, 2010; cited in the manuscript) that the higher susceptibility of hypomorphic ADAM17 mice was related to reduced shedding of EGF-R ligands in this model. Therefore, the authors should address shedding of these ligands in Frmd8/iTAP knock-out mice. **Response at time of initial submission.** We agree that it would be interesting to determine whether the enhanced susceptibility is related to reduced shedding of EGFR ligands. Chalaris et al. (Chalaris et al., 2010) showed (Figure 4A) that the overall levels of the EGFR ligand TGF α was elevated in the colons of ADAM17^{ex/ex} mice when exposed to DSS. We will carry out a similar IHC staining for TGF α in our model, as well as imaging Ki67 as a marker for cellular proliferation associated with epithelial repair. This will complement our data (Fig. 5E,F) that show the defective shedding of the endogenous EGFR ligand AREG from mouse Lewis lung carcinoma cells.

Updated Response. To determine whether enhanced susceptibility of iTAP/Frmd8 KO mice to DSS-induced colitis was due to reduced shedding of the EGFR ligand TGF- α , we aimed to perform TGF- α immunohistochemistry analysis on colon sections from WT versus iTAP/Frmd8 KOs following exposure to the DSS model. We aimed to use the antibody that was described in the work of Chalaris and colleagues (Chalaris et al., 2010) but since this study was carried out ~12 years' ago, it was not possible to source the original commercial antibody. The authors were unfortunately unable to recommend to us a currently-available antibody that would be suitable for the detection of TGF- α by IHC in mouse tissues. We therefore tested five different commercially-sourced TGF- α antibodies selected from reliable companies. Unfortunately, relative to control tissue sections stained with secondary antibodies alone, none of these appears to be specific, to our experienced histopathology facility, in controlled trial experiments in a range of tissues selected because of their established high expression of TGF- α . The technical teams of these companies were unable to help us with alternative reagents (beyond those described below), troubleshooting or alternative protocols. We conclude that unfortunately we cannot perform this experiment to detect TGF- α with confidence.

Summary of anti-TGF- α antibodies used:

1. **Rabbit anti-TGF- α antibody (Abcam, ab9585, 1:200).** For a positive control we used skin, a tissue that expresses high levels of TGF- α (Finzi et al., 1991). The staining was inconsistent, and did not stain all of the keratinocyte population as anticipated (Gottlieb et al., 1988) (data not shown). We also tested the liver, as suggested by Abcam, but it non-specifically stained all liver structures (data not shown). After our complaint, Abcam removed IHC-P as a recommended application from the datasheet of the antibody online.

2. **Recombinant anti-TGF- α antibody (Abcam, ab208156, 1:100).** In response to the lack of specificity of the first antibody used, the Abcam technical team suggested this antibody instead, which is claimed to be suitable for IHC-P. On mouse skin, the antibody was inconsistent (stained only a proportion of keratinocytes data not shown), contrary to expectations (Gottlieb *et al.*, 1988). In the liver there was no positive staining (data not shown).
3. **Rabbit anti-TGF- α antibody (ThermoFisher Scientific, PA5-102314, 1:50).** When tested in skin, we found unspecific staining with reactivity in fibroblasts, immune cells, endothelial cells and in adipocytes in addition to the anticipated epidermal staining (data not shown). We also found a strong non-specific signal inside the nuclei of multiple cell types. To confirm in our organ of interest, we tested the colon, where the antibody stained both the cytoplasm and nuclei through all the intestinal epithelium, suggesting that the staining was not specific for TGF- α (Fig. 1A,B).
4. **Rabbit anti-TGF- α antibodies (ThermoFisher Scientific, PA5-104226, 1:50 and ThermoFisher Scientific, PA5-86213, 1:50).** For a positive control we used skin, where there was no positive staining for either antibody above the background staining associated with the secondary antibody only (data not shown). We also tested staining in mouse lung carcinoma sections, which should have significant expression of TGF- α (Liu *et al.*, 1990), but no specific signal was detected.

Figure R2. IHC staining of paraffin-embedded mouse colon samples with anti-TGF- α antibodies

A-B. Non-specific TGF- α IHC staining (with pronounced nuclear staining) in sections from a WT mouse colon. **(A)** A negative control where the sample was incubated only with the secondary antibody; **(B)** staining with a rabbit anti-TGF- α primary antibody (ThermoFisher Scientific, PA5-102314, 1:50). Antigen retrieval was carried out using sodium citrate buffer for 50 min at 99 °C, permeabilization with PBS-Tween 20, and blocking with Protein Block serum (X0909, Dako) for 60 min. The secondary antibody used was EnVision HRP labelled polymer anti-rabbit (K4002, Dako) for 30 min and for detection DAB (K3468, Dako) was used. Whole slide images were acquired with Hamamatsu Nanozoomer slide scanner. A scale bar of 100 μ m is indicated within the IHC images.

We also attempted Ki67 staining on colon samples. However, since the samples were embedded in a “swiss roll” conformation to allow a proper histopathological evaluation of the whole large intestine, this makes it impossible to control the orientation of each part of the colon. Consequently, some intestinal sections will contain areas corresponding to more basal regions (where proliferating stem cells are located in the intestinal crypts) than others. As the orientation of the sections are a key determinant for a standardized and correct assessment in the other assays, and given the additional time and number of animals that would be required to repeat this *de novo*, unfortunately we are unable to carry out this analysis.

Nonetheless, to better understand whether the increased susceptibility of KO mice to colitis is related to defects in epithelial repair (see comments and response to reviewer 2, below), we analyzed the ability of the animals to recover body weight following withdrawal of DSS. We administrated 2 % DSS in the drinking water for 5 days, and afterwards DSS was replaced by water, while monitoring the mice for recovery until euthanasia at day 10. Body weight was measured 3 times per week and the colon length was measured. This experiment

indicates that iTAP/Frmd8 KO mice are less efficient at recovering body weight than the WT mice (**now Supplementary Fig. 5B**), had a shorter colon (**now Supplementary Fig. 5C**), and had more severe IBD as judged by histopathological analysis, specifically in the mid colon (**now Supplementary Fig. 5D-F**). These new data support a role for iTAP/Frmd8 in the promotion of intestinal epithelial repair after DSS-induced damage. We also note our data from the original version of the MS (**Fig. 5E,F**) that show the defective shedding of Amphiregulin from iTAP/Frmd8 KO Lewis lung carcinoma cells, and therefore demonstrates that iTAP/Frmd8 is needed for ADAM17-associated shedding of EGFR ligands.

6. In the experiment shown in Fig. 5, the authors inject parental and Frmd8/iTAP knock-out LLC tumor cells into WT mice. The note that the number of metastases, tumor volume and tumor burden is dramatically decreased. In the study by Bolik et al, 2022 (cited in the manuscript) it has been shown that in hypomorphic ADAM17 mice, metastasis formation by LLC tumor cells was dramatically reduced. In this study it was also shown that ADAM17 activity in endothelial cells was responsible for this effect, which was at least in part mediated by TNF-RI and TNF-RII. This mechanistic difference should be addressed in the manuscript.

7. Along the same line: when tumor cells are injected i.v., the cells need to extravasate before they can form tumors. The authors need to mechanistically address whether the effects of Frmd8/iTAP are on extravasation or on tumor growth (or both). **Response at time of initial submission.** The reviewer addresses an interesting question: whether iTAP expression in endothelial cells modulates tumor extravasation and mechanistically whether this depends on the TNF signaling pathway. We argue that it this would be better mechanistically addressed in cell culture models similar to those used by Bolik et al. (Bolik et al., 2022). We will assess the transwell migration of tumor cells through monolayers of endothelial cells in which Frmd8/iTAP expression is ablated. The dependency on the TNF-associated endothelial cell necroptosis will also be addressed. In addition, the experiments mentioned in the response to the point made by reviewer No. 2 (below) has now addressed tumor proliferation more thoroughly.

Updated Response. Bolik and colleagues showed elegantly and thoroughly that deletion of ADAM17 in the endothelium reduced endothelial permeability, which inhibits tumor cells extravasation and consequently metastasis. They also discovered that ADAM17-mediated shedding of TNFR1 in endothelial cells is required for tumor cell-induced endothelial cell death, to facilitate tumor cell extravasation. As iTAP/Frmd8 is important for ADAM17 activity, the hypothesis is that the absence of iTAP/Frmd8 in the endothelium would have a similar effect as ADAM17 loss (e.g. to render the endothelium less permeable to tumor cells). Unfortunately, as our iTAP/Frmd8 model is a global KO, the results obtained (protection from tumor growth, **Fig. 4B-E**) cannot easily be attributed to the role of iTAP/Frmd8 in the endothelium. In this case, the effects could be caused by ADAM17 shedding defects within the tumor microenvironment (e.g., endothelial cells, immune cells, fibroblasts as well as the extracellular matrix and signaling molecules from the recipient WT mouse host). We believe that the role of iTAP/Frmd8 in the endothelium on endothelial necroptosis and tumor extravasation should be addressed, but requires the generation of endothelial-specific Frmd8/iTAP KO mice, which are not yet available.

Nevertheless, we attempted to carry out the protocol described by Bolik and colleagues, to evaluate the transwell migration of MDA-MB231 tumor cells through monolayers of HUVEC endothelial cells in which iTAP/Frmd8 expression was ablated using siRNA. However, from preliminary experiments we realized that these are technically demanding assays; in particular, we are not confident in our ability to discriminate whether a CFSE-labelled MDA-MB231 tumor cell in our images is on the lower side of the transwell (i.e., has successfully passed through the HUVEC monolayer) or still remains on the other side of the transwell (i.e. where the HUVECs are). Given the time constraints and as reproducing these experiments with iTAP/Frmd8 would not advance us conceptually beyond the impressive work of Bolik and colleagues, we prefer not to pursue these experiments further.

Finally, the reviewer also asked whether the impact of iTAP/Frmd8 was upon tumor growth. We have been able to confirm that iTAP/Frmd8 expression is important for the proliferative capacity of the tumor cells in vivo (by quantifying Ki67 IF positive cells in tumor sections of iTAP KO and OE versus respective controls, **Fig. 5K-L**, and by quantifying the mRNA level of the proliferative marker PCNA on those tumors, **Supplementary Fig. 8A**). Therefore, we can conclude that iTAP/Frmd8 can influence tumor cell proliferation.

Minor points:

1. The authors name the protein Frmd8/iTAP sometimes as Frmd8 and sometimes as iTAP. This is confusing for

the reader. Since the protein has been characterized under both names, the authors should stick to Frmd8/iTAP. **Response:** now corrected throughout, to iTAP/Frmd8.

2. Along the same line: the authors should stick to the name ADAM17 and not sometimes switch to the older name TACE. **Response:** now corrected throughout.

3. The authors use Frmd8/iTAP knock-out mice. It is not clear from the statement of p5, whether they use the mice described in Künzel et al, 2018 or the mice described in Oikonomidi et al, 2018. This should be clarified. **Response:** now corrected-we used the mice generated by our group, described in Oikonomidi et al., 2018.

4. Some references (e.g. Dong et al, 1999 and Gschwind et al, 2003) are incomplete. **Response:** now corrected throughout.

Response to Reviewer #2

Reviewer #2 notes that “This work adds additional data to support the importance of iTAP/sheddase complex/ADAM17 in disease development. Most importantly, it suggests a role for iTAP in tumor progression.” We thank the reviewer for their insightful critiques and suggestions, which have helped us to improve our MS substantially.

Figure 1: The findings regarding L-selectin shedding are very clear and perhaps meaningful. However, a discussion putting these findings in context with the disease models used later in the manuscript is warranted. Currently, inclusion of these data does not add much to the story if not discussed or referenced later in the manuscript. **Response.** We agree that these data are striking and that they were not properly placed within context in the paper. These data have now been discussed more extensively in the Results and Discussion sections of the revised version of the paper.

We have added the following new text:

“L-selectin is a prominent cell adhesion molecule expressed on most leukocytes and plays a role in the migration of neutrophils and other immune cells to sites of inflammation (Ivetic et al., 2019). L-selectin is shed upon immune cell activation; ADAM17 appears to be the major L-selectin-shedding protease since thymocytes and other leukocytes lacking ADAM17 fail to shed L-selectin upon their activation by the phorbol ester PMA (Peschon et al., 1998; Li et al., 2006).”

“Shedding defects have been reported for iRhom2 mutant mice, which also develop normally, but exhibit pronounced inflammatory impairments and are defective in the shedding of TNF and L-selectin (Adrain et al., 2012; McIlwain et al., 2012; Siggs et al., 2012). Notably, ADAM17-deficient neutrophils exhibit alterations in rolling velocity and adhesion in response to inflammatory insults that can be normalized with L-selectin-neutralizing antibodies (Tang et al., 2011). In addition, there is a positive correlation between circulating soluble L-selectin levels and disease severity in human Ulcerative Colitis patients (Seidelin et al., 1998). Therefore, future studies on the role of iTAP/Frmd8 in inflammatory models, particularly those associated with leukocyte infiltration are warranted.”

Figure 2: The authors observe that iTAP KO mice have worse outcomes following DSS-colitis. In the text, they mention that iRhom2 KO mice do not phenocopy the iTAP KO mice following DSS-colitis, yet no explanation is offered. If the mechanism of iTAP is proposed to be through iRhom2 activity and ADAM17 shedding, you would expect the iRhom2 KOs to demonstrate similar intestinal phenotypes. The authors should comment on this discrepancy. **Response:** We apologize for not proposing a potential explanation for this in the original version. Our published data (Christova et al., 2013) show that iRhom1 is expressed alongside iRhom2 in the large intestine. Hence, iRhom1 levels may be sufficient to prevent ADAM17-associated defects in iRhom2 KO mice in response to experimental colitis. We have adjusted the narrative in the Discussion section to note this potential explanation.

We have added the following new text:

“Interestingly, unlike our observed sensitization of iTAP/Frmd8 KO mice to DSS-triggered colitis (Fig.3, Supplemental Fig.5), it has been shown that iRhom2 KO mice are not sensitized to DSS-induced colitis (Siggs et al., 2012; Geesala et al., 2019). A potential explanation for this discrepancy is that iRhom1, which plays a redundant role with iRhom2 in ADAM17 regulation, is expressed alongside iRhom2 in the intestinal tract, including the large intestine (Christova et al., 2013). iRhom1 expression may mask the impact of iRhom2 deletion, explaining the lack of sensitization of iRhom2 KO mice to DSS-induced colitis (Siggs et al., 2012; Geesala et al., 2019).”

Additionally, the conclusion about the importance of iTAP in intestinal repair would be better supported if the DSS colitis experiments were continued to later time points to include the recovery phase (once the mice return to original body weight), rather than just ending the experiment at peak repair. **Response:** Unfortunately, our animal license does not permit additional permutations in the protocol (e.g. extending beyond the timepoints post-DSS that we currently use). Nonetheless, as this experimental modification is predicted not to surpass moderate severity limits, we have queried with our institute’s ethics committee whether permission could be sought to do this. We await the response. However, we argue that the current experimental setup includes a component of recovery, since the animals have been allowed to continue for 2 days post-DSS exposure. We believe that the experiments that will assess whether TGFA levels accumulate in tissue sections from DSS-treated mice, as well as the assessment of whether iTAP KO colonic epithelia undergo reduced proliferation, will help to strengthen the premise that the colonic phenotype is associated with defective EGFR-associated epithelial repair.

Updated Response: We have been able to carry out this experiment (see response to Reviewer#1 on page 4) which is now incorporated in **Supplementary Fig. 5**. In summary, WT mice recover body weight post-DSS withdrawal much more efficiently than iTAP/Frmd8 KOs (**Supplementary Fig. 5B**). In addition, the KOs present a shorter colon (**Supplementary Fig. 5C**), and worse IBD histopathological score, mainly in the mid colon (**Supplementary Fig. 5D-F**). Therefore, our new data support a role for iTAP/Frmd8 in the promotion of intestinal epithelial repair after DSS-induced damage.

We have added the following new text:

“To address the effect of iTAP/Frmd8 loss on epithelial repair capacity after the DSS insult, we induced colitis but let the mice recover for 5 days after DSS exposure (Supplementary Fig. 5A). This established that the KO mice could not recover body weight to the same extent as the WT mice (Supplementary Fig. 5B), had a shorter colon (Supplementary Fig. 5C), and had more severe IBD in the histopathological analysis, specifically in the mid colon (Supplementary Fig. 5D-F). Our data hence establish that like ADAM17 loss of iTAP/Frmd8 exacerbates experimental colitis in vivo and impairs the capacity of recovery after the insult.”

Figure 3: The authors make the statement that "...although inflammatory infiltrates were modest in the lungs of mice..." Is this based on histology alone? If the authors want to make this claim, they must assess immune infiltrates directly (e.g. using flow cytometry). **Response:** This is a fair point—it was based on histology alone. We have now removed this sentence.

The authors evaluate lung metastasis in the LLC subcutaneous model but any conclusion about metastasis cannot be made in this model without looking at primary tumors of a similar size. Metastasis tends to be associated with the size of the primary tumor so smaller primary tumors usually mean lower levels of metastasis (without being able to parse apart direct effects on the metastatic process). I assume that the data in Fig 3H are from mice with different tumor sizes--in order to properly evaluate this, the authors need to euthanize WT and KO animals with similar tumor burdens and compare metastatic burden. **Response:** We thank the reviewer for pointing this out. We agree and have now removed the data in original Fig. 3H and the text associated with this.

Including total mRNA levels of cytokines does not add to this figure. First, bulk levels of mRNA are not a good way to evaluate the state of a tumor (immune cell phenotype/activity would be better). Second, TNF and IL-6 were used in previous figures as readouts of ADAM17 activity (or not) and here are just markers of inflammation? This is confusing/contradictory. If included, this should be moved to the supplement.

Response: Yes, the mRNA levels of TNF and IL6 were used as general readouts for inflammation here. We agree that, in our context, this could be confusing, while relying elsewhere on shedding of the TNF protein as a readout for ADAM17 activity. We have moved these mRNA-based data into **Supplementary Fig. 6A,B** to minimize confusion.

Figure 4: Claims about proliferation cannot be made here because the results as shown are not significant (Fig 4K) [now current Fig. 5K]. Additional readouts for proliferation should be used to support this conclusion.

Response: We agree and have now added IF images showing Ki67 expression (**Fig. 5K,L**) plus data showing the mRNA levels of the proliferative marker PCNA (**Supplementary Fig. 8A**).

Similar to Figure 3, claims about metastasis cannot be made from these experiments without comparing mice with similar primary tumor burden. The metastasis data in Figure 5 are much more solid and convincing.

Response: We agree and have removed the relevant data (**Fig. 3H,I** and **Supplementary Fig. 5**) and the associated text.

Figure 5: The authors use Fig 5 K & L as evidence that tumor cells proliferated more or less rapidly, depending on expression levels of iTAP. The data do not support this statement. If I understand the methods correctly, this assay involves plating of 500K tumor cells and then harvesting after several days. Upon harvest, there were 100 fold fewer cells (~5K). To me this indicates effects on survival, not proliferation. Proliferation was never measured in this assay. Without these data, the authors can make no claim regarding the mechanisms of tumor cell autonomous functions of iTAP.

Response: We apologize for the ambiguity in how we described this experiment. The assay involved plating 500,000 tumor cells per well. After approximately 72h, the cells were trypsinized and an aliquot of each cell suspension was supplemented with a defined volume of counting beads. The cell density in each sample, normalized against a defined number of counting beads (500) was determined by FACS. Hence, the readout is the relative cell density between individual wells (e.g. proliferation) and is not, as aberrantly stated an indication of the absolute number of cells in the wells. We have now clarified this in the relevant figure, figure legends and materials and methods.

Minor comments:

The language regarding any results that are not statistically-significant need to be softened in the text. In several places, there are statements about non-significant results that are much too definitive and somewhat misleading. Non statistically-significant results can be useful to include to show trends (as in Fig 5 G-I), but the interpretation should not be overstated.

Response: Our apologies that we did not get the balance right. We hope that we have now toned the relevant comments down.

The title is overstated. In this manuscript, the authors do not show clear mechanistic links for iTAP promoting epithelial repair (worse outcomes after DSS are not just caused by decreased repair). The strongest data in the manuscript are those regarding tumor growth. This should be highlighted in the title.

Response: Thanks for the suggestion. We have now modified the title, to remove references to epithelial repair and to emphasize the impact on tumor growth. We have also added the experiment (**Supplementary Fig. 5**) that shows that iTAP/Frmd8 KO mice are defective in recovery from DSS-induced colitis (see above).

Response to Reviewer #3

Reviewer #3. Notes that “ This study is exciting. It shows for the first time the pathophysiological role of iTAP in vivo and has major implications for ADAM17, which is a drug target in numerous diseases, in particular sepsis, inflammation and tumors. This study is an important step towards the use of iTAP as a drug target. Thus, this study will be of interest to basic scientists studying ADAM17, its regulation, its substrate specificity and its physiological functions. The study will also be of interest to translational scientists in academia and

pharma/biotech studying the numerous ADAM17-dependent diseases. A clear strength of the study is the inclusion of different disease models, where iTAP plays a role (protective or non-protective), and the demonstration that iTAP contributes to tumors both in the tumor niche and in the tumor itself.”

Response: We thank the reviewer for their supportive and insightful critiques. We agree that our work opens exciting prospects for the investigation of iTAP/Frmd8 as a drug target.

My only major concern is the choice of the control LLC cells. The cells used are not the ideal control for the iTAP knock-out cells. Both wild-type and iTAP knock-out cells were transduced with a Cas9 expressing vector, as it should be. But only the knock-out cells were further transduced with a virus expressing the gRNAs against iTAP, whereas the control cells apparently were not transduced with a virus expressing control gRNAs. Three single cell clones of the iTAP ko cells were pooled for in vivo injection, whereas the parental pool (and not clones) were apparently used as a control. My concern is that the ko cells do not only differ from the wild-type cells due to the knock-out of iTAP, but potentially also due to other gene expression alterations resulting from the additional transduction of the knock-out cells with a gRNA virus and because of the selection of single cell clones. Such expression changes beyond the simple lack of iTAP may have a major influence on those tumor phenotypes in vivo, where these cells were used. Ideally, the authors would generate an additional, independent pool of iTAP knock-out cells and repeat one of the crucial in vivo experiments. As a time-saving alternative, the authors need to demonstrate that the iTAP knock-out cells are nearly identical to the control cells (with the exception of iTAP). This could be done by RNA sequencing or cell lysate proteomics or by blotting for several different proteins (at least 10 from different compartments) and demonstrating that there is no significant change in protein abundance - apart from iTAP.

Response: Thank you for raising this concern. While the *in vivo* experiments used Cas9-expressing cells as the control, the equivalent *in vitro* experiments used the Cas9-expressing cells further transduced to express the empty vector, which we agree is a better control. As we observe analogous results compared to their respective controls in the *in vivo* setting compared to the *in vitro* setting, we are confident that these results are genuine. Moreover, our *in vivo* and *in vitro* experiments that compare vector versus Frmd8/iTAP-overexpressing cells show the anticipated opposite effect compared to the KO cells, bolstering our confidence in these observations. Nonetheless, as suggested to address the issue of potential differences in protein abundance, we now provide western blot images and their associated densitometric analysis showing that the expression levels of 13 different proteins from a range of compartments are unaltered (**Supplementary Fig 7A-M**).

Minor points.

1. Indicate the concentrations of the used drugs (marimastat, PMA) in the figure legends.

Response: Apologies for this omission, which we have now corrected.

2. Indicate in the manuscript that LLC cells are of mouse origin. **Response:** Apologies for this omission, which was now incorporated.

3. Page 6, top paragraph: it is not clear to me, whether there is an eye phenotype or not. Please rephrase this sentence. **Response:** we have corrected this to make it clearer that the iTAP/Frmd8 KO eyes develop normally (as per WT animals).

We have added the following new text:

“During mouse embryonic development, the eyelids of WT mice undergo an ADAM17/EGFR-dependent epithelial fusion step (Peschon et al., 1998). WT mice are born with the appearance of their eyes “closed”, after which eyelids open 11-12 days after birth. By contrast, ADAM17-deficient embryos are born with their eyes precociously open because of a failure in epithelial eyelid fusion (Peschon et al., 1998). Like iRhom2 KOs, whose eyelids develop normally, probably because of redundancy with iRhom1, iTAP/Frmd8 KOs lacked ADAM17-associated defects in the eye, which were indistinguishable from WT eyes at E14.5 (Fig. S1B) and after birth. Other epithelial tissues (small intestine, colon, caecum, lung) and the heart (Fig. S1C,D, respectively) also exhibited no obvious differences

compared to controls. We conclude that *iTAP/Frmd8*-null mice undergo normal epithelial and cardiac development.”

4. Figure legend 1: "...with 3 replicates per experiment". Indicate whether this refers to biological or technical replicates. Response: we have corrected this-they were technical replicates.

5. Indicate in figure legends which statistical test was used. Response: we have now indicated this throughout.

6. Fig. 2F [current figure 3F]. The y-axis label should be body weight and not body weight loss. Response: now corrected.

7. Fig. 4C: the increase in the 75 kDa fragment upon *iTAP* OE is difficult to see. Can you quantify the increase? And also the reduction in the KO cells? Response: we have now incorporated this quantification into the revised MS (Supplementary Fig 7N-O).

References cited

- Adrain C, Zettl M, Christova Y, Taylor N, Freeman M (2012) Tumor necrosis factor signaling requires iRhom2 to promote trafficking and activation of TACE. *Science*, **335**: 225–228
- Bolik J, Krause F, Stevanovic M, Gandraß M, Thomsen I, Schacht SS, Rieser E, Müller M, Schumacher N, Fritsch J, Wichert R, Galun E, Bergmann J, Röder C, Schafmayer C, Egberts JH, Becker-Pauly C, Saftig P, Lucius R, Schneider-Brachert W, Barikbin R, Adam D, Voss M, Hitzl W, Krüger A, Strilic B, Sagi I, Walczak H, Rose-John S, Schmidt-Arras D (2022) Inhibition of ADAM17 impairs endothelial cell necroptosis and blocks metastasis. *J Exp Med*, **219**: e20201039
- Cavadas M, Oikonomidi I, Gaspar CJ, Burbridge E, Badenes M, Félix I, Bolado A, Hu T, Bileck A, Gerner C, Domingos PM, von Kriegsheim A, Adrain C (2017) Phosphorylation of iRhom2 Controls Stimulated Proteolytic Shedding by the Metalloprotease ADAM17/TACE. *Cell Rep*, **21**: 745–757
- Chalaris A, Adam N, Sina C, Rosenstiel P, Lehmann-Koch J, Schirmacher P, Hartmann D, Cichy J, Gavrilova O, Schreiber S, Jostock T, Matthews V, Häsler R, Becker C, Neurath MF, Reiss K, Saftig P, Scheller J, Rose-John S (2010) Critical role of the disintegrin metalloprotease ADAM17 for intestinal inflammation and regeneration in mice. *J Exp Med*, **207**: 1617–1624
- Christova Y, Adrain C, Bambrough P, Ibrahim A, Freeman M (2013) Mammalian iRhoms have distinct physiological functions including an essential role in TACE regulation. *EMBO Rep*, **14**: 884–890
- Finzi E, Harkins R, Horn T (1991) TGF- α is widely expressed in differentiated as well as hyperproliferative skin epithelium. *J Invest Dermatol*, **96**: 328–332
- Geesala R, Schanz W, Biggs M, Dixit G, Skurski J, Gurung P, Meyerholz DK, Elliott D, Issuree PD, Maretzky T (2019) Loss of RHBDF2 results in an early-onset spontaneous murine colitis. *J Leukoc Biol*, **105**: 767–781
- Gottlieb ALICEB, Chang CHEOWKHONG, Posnett DAVIDN, Fanelli B, Tam JAMESP (1988) Detection of transforming growth factor alpha in normal, malignant, and hyperproliferative human keratinocytes. *The Journal of experimental medicine*, **167**: 670–675
- Ivetic A, Green HLH, Hart SJ (2019) L-selectin: a major regulator of leukocyte adhesion, migration and signaling. *Frontiers in immunology*,
- Li X, Maretzky T, Weskamp G, Monette S, Qing X, Issuree PD, Crawford HC, McIlwain DR, Mak TW, Salmon JE, Blobel CP (2015) iRhoms 1 and 2 are essential upstream regulators of ADAM17-dependent EGFR signaling. *Proc Natl Acad Sci U S A*, **112**: 6080–6085
- Li Y, Brazzell J, Herrera A, Walcheck B (2006) ADAM17 deficiency by mature neutrophils has differential effects on L-selectin shedding. *Blood*, **108**: 2275–2279
- Liu C, Woo A, Tsao MS (1990) Expression of transforming growth factor- α in primary human colon and lung carcinomas. *British journal of cancer*, **62**: 425–429
- McIlwain DR, Lang PA, Maretzky T, Hamada K, Ohishi K, Maney SK, Berger T, Murthy A, Duncan G, Xu HC, Lang KS, Häussinger D, Wakeham A, Itie-Youten A, Khokha R, Ohashi PS, Blobel CP, Mak TW (2012) iRhom2 regulation of TACE controls TNF-mediated protection against *Listeria* and responses to LPS. *Science*, **335**: 229–232

- Oikonomidi I, Burbridge E, Cavadas M, Sullivan G, Collis B, Naegele H, Clancy D, Brezinova J, Hu T, Bileck A, Gerner C, Bolado A, von Kriegsheim A, Martin SJ, Steinberg F, Strisovsky K, Adrain C (2018) iTAP, a novel iRhom interactor, controls TNF secretion by policing the stability of iRhom/TACE. *Elife*, **7**: e35032
- Peschon JJ, Slack JL, Reddy P, Stocking KL, Sunnarborg SW, Lee DC, Russell WE, Castner BJ, Johnson RS, Fitzner JN, Boyce RW, Nelson N, Kozlosky CJ, Wolfson MF, Rauch CT, Cerretti DP, Paxton RJ, March CJ, Black RA (1998) An essential role for ectodomain shedding in mammalian development. *Science*, **282**: 1281–1284
- Seidelin JB, Vainer B, Horn T, Nielsen OH (1998) Circulating L-selectin levels and endothelial CD34 expression in inflammatory bowel disease. *The American journal of gastroenterology*, **93**: 1854–1859
- Siggs OM, Xiao N, Wang Y, Shi H, Tomisato W, Li X, Xia Y, Beutler B (2012) iRhom2 is required for the secretion of mouse TNF α . *Blood*, **119**: 5769–5771
- Tang J, Zarbock A, Gomez I, Wilson CL, Lefort CT, Stadtmann A, Bell B, Huang LC, Ley K, Raines EW (2011) Adam17-dependent shedding limits early neutrophil influx but does not alter early monocyte recruitment to inflammatory sites. *Blood*, **118**: 786–794

December 15, 2022

RE: Life Science Alliance Manuscript #LSA-2022-01644R

Dr. Colin Adrain
Gulbenkian
Rua da Quinta Grande 6
Oeiras 2780-156
Portugal

Dear Dr. Adrain,

Thank you for submitting your revised manuscript entitled "The ADAM17 sheddase complex regulator iTAP/Frmd8 modulates inflammation and tumor growth". We would be happy to publish your paper in Life Science Alliance pending final revisions necessary to meet our formatting guidelines.

- please upload your main manuscript text as an editable doc file
- please upload both your main and supplementary figures as single files
- please make sure that the author order in the manuscript matches with the author order in our system
- please add the author contributions to the main manuscript
- please use the [10 author names, et al.] format in your references (i.e. limit the author names to the first 10)
- please remove the Data Availability statement since there is nothing specific to mention here

A. FINAL FILES:

B. MANUSCRIPT ORGANIZATION AND FORMATTING:

**Submission of a paper that does not conform to Life Science Alliance guidelines will delay the acceptance of your

manuscript.**

The license to publish form must be signed before your manuscript can be sent to production. A link to the electronic license to publish form will be sent to the corresponding author only. Please take a moment to check your funder requirements.

Sincerely,

January 3, 2023

RE: Life Science Alliance Manuscript #LSA-2022-01644RR

Dr. Colin Adrain
Gulbenkian
Rua da Quinta Grande 6
Oeiras 2780-156
Portugal

Dear Dr. Adrain,

Thank you for submitting your Research Article entitled "The ADAM17 sheddase complex regulator iTAP/Frmd8 modulates inflammation and tumor growth". It is a pleasure to let you know that your manuscript is now accepted for publication in Life Science Alliance. Congratulations on this interesting work.

DISTRIBUTION OF MATERIALS:

Again, congratulations on a very nice paper. I hope you found the review process to be constructive and are pleased with how the manuscript was handled editorially. We look forward to future exciting submissions from your lab.

Sincerely,
